# HiPPO: Recurrent Memory with Optimal Polynomial Projections

**Albert Gu**[*][†]**, Tri Dao**[*][†]**, Stefano Ermon** [†]**, Atri Rudra** [‡]**, Christopher Ré** [†]
[†] Department of Computer Science, Stanford University
[‡] Department of Computer Science and Engineering, University at Buffalo, SUNY
{albertgu,trid}@stanford.edu, ermon@cs.stanford.edu, atri@buffalo.edu, chrismre@cs.stanford.edu

## Abstract

A central problem in learning from sequential data is representing cumulative history in an incremental fashion as more data is processed. We introduce a general framework (HiPPO) for the online compression of continuous signals and discrete time series by projection onto polynomial bases. Given a measure that specifies the importance of each time step in the past, HiPPO produces an optimal solution to a natural *online function approximation* problem. As special cases, our framework yields a short derivation of the recent Legendre Memory Unit (LMU) from first principles, and generalizes the ubiquitous gating mechanism of recurrent neural networks such as GRUs. This formal framework yields a new memory update mechanism (HiPPO-LegS) that scales through time to remember all history, avoiding priors on the timescale. HiPPO-LegS enjoys the theoretical benefits of timescale robustness, fast updates, and bounded gradients. By incorporating the memory dynamics into recurrent neural networks, HiPPO RNNs can empirically capture complex temporal dependencies. On the benchmark permuted MNIST dataset, HiPPO-LegS sets a new state-of-the-art accuracy of 98.3%. Finally, on a novel trajectory classification task testing robustness to out-of-distribution timescales and missing data, HiPPO-LegS outperforms RNN and neural ODE baselines by 25-40% accuracy.

## 1 Introduction

Modeling and learning from sequential data is a fundamental problem in modern machine learning, underlying tasks such as language modeling, speech recognition, video processing, and reinforcement learning. A core aspect of modeling long-term and complex temporal dependencies is *memory*, or storing and incorporating information from previous time steps. The challenge is learning a representation of the entire cumulative history using bounded storage, which must be updated online as more data is received.

One established approach is to model a state that evolves over time as it incorporates more information. The deep learning instantiation of this approach is the recurrent neural network (RNN), which is known to suffer from a limited memory horizon [34, 38, 56] (e.g., the "vanishing gradients" problem). Although various heuristics have been proposed to overcome this, such as gates in the successful LSTM and GRU [16, 34], or higher-order frequencies in the recent Fourier Recurrent Unit [79] and Legendre Memory Unit (LMU) [71], a unified understanding of memory remains a challenge. Furthermore, existing methods generally require priors on the sequence length or timescale and are ineffective outside this range [66, 71]; this can be problematic in settings with distribution shift (e.g. arising from different instrument sampling rates in medical data [62, 63]). Finally, many of them lack theoretical guarantees on how well they capture long-term dependencies, such as gradient bounds. To design

---

[*]Equal contribution. Order determined by coin flip.

a better memory representation, we would ideally (i) have a unified view of these existing methods, (ii) be able to address dependencies of any length without priors on the timescale, and (iii) have a rigorous theoretical understanding of their memory mechanism.

Our insight is to phrase *memory* as a technical problem of *online function approximation* where a function $f(t) : \mathbb{R}_+ \to \mathbb{R}$ is summarized by storing its optimal coefficients in terms of some basis functions. This approximation is evaluated with respect to a measure that specifies the importance of each time in the past. Given this function approximation formulation, orthogonal polynomials (OPs) emerge as a natural basis since their optimal coefficients can be expressed in closed form [14]. With their rich and well-studied history [65], along with their widespread use in approximation theory [68] and signal processing [57], OPs bring a library of techniques to this memory representation problem. We formalize a framework, **HiPPO** (high-order polynomial projection operators), which produces operators that project arbitrary functions onto the space of orthogonal polynomials with respect to a given measure. This general framework allows us to analyze several families of measures, where this operator, as a closed-form ODE or linear recurrence, allows fast incremental updating of the optimal polynomial approximation as the input function is revealed through time.

By posing a formal optimization problem underlying recurrent sequence models, the HiPPO framework (Section 2) generalizes and explains previous methods, unlocks new methods appropriate for sequential data at different timescales, and comes with several theoretical guarantees. (i) For example, with a short derivation we exactly recover as a special case the LMU [71] (Section 2.3), which proposes an update rule that projects onto fixed-length sliding windows through time.[2] HiPPO also sheds new light on classic techniques such as the gating mechanism of LSTMs and GRUs, which arise in one extreme using only low-order degrees in the approximation (Section 2.5). (ii) By choosing more suitable measures, HiPPO yields a novel mechanism (Scaled Legendre, or LegS) that always takes into account the function's full history instead of a sliding window. This flexibility removes the need for hyperparameters or priors on the sequence length, allowing LegS to generalize to different input timescales. (iii) The connections to dynamical systems and approximation theory allows us to show several theoretical benefits of HiPPO-LegS: invariance to input timescale, asymptotically more efficient updates, and bounds on gradient flow and approximation error (Section 3).

We integrate the HiPPO memory mechanisms into RNNs, and empirically show that they outperform baselines on standard tasks used to benchmark long-term dependencies. On the permuted MNIST dataset, our hyperparameter-free HiPPO-LegS method achieves a new state-of-the-art accuracy of 98.3%, beating the previous RNN SoTA by over 1 point and even outperforming models with global context such as transformers (Section 4.1). Next, we demonstrate the timescale robustness of HiPPO-LegS on a novel trajectory classification task, where it is able to generalize to unseen timescales and handle missing data whereas RNN and neural ODE baselines fail (Section 4.2). Finally, we validate HiPPO's theory, including computational efficiency and scalability, allowing fast and accurate online function reconstruction over millions of time steps (Section 4.3). Code for reproducing our experiments is available at `https://github.com/HazyResearch/hippo-code`.

## 2 The HiPPO Framework: High-order Polynomial Projection Operators

We motivate the problem of online function approximation with projections as an approach to learning memory representations (Section 2.1). Section 2.2 describes the general HiPPO framework to derive memory updates, including a precise definition of the technical problem we introduce, and an overview of our approach to solving it. Section 2.3 instantiates the framework to recover the LMU and yield new memory updates (e.g. HiPPO-LagT), demonstrating the generality of the HiPPO framework. Section 2.4 discusses how to convert the main continuous-time results into practical discrete versions. Finally in Section 2.5 we show how gating in RNNs is an instance of HiPPO memory.

### 2.1 HiPPO Problem Setup

Given an input function $f(t) \in \mathbb{R}$ on $t \geq 0$, many problems require operating on the cumulative *history* $f_{\leq t} := f(x)\,|_{x \leq t}$ at every time $t \geq 0$, in order to understand the inputs seen so far and make future predictions. Since the space of functions is intractably large, the history cannot be perfectly memorized

and must be compressed; we propose the general approach of projecting it onto a subspace of bounded dimension. Thus, our goal is to maintain (online) this compressed representation of the history. In order to specify this problem fully, we require two ingredients: a way to quantify the approximation, and a suitable subspace.

**Function Approximation with respect to a Measure.**   Assessing the quality of an approximation requires defining a distance in function space. Any probability measure $\mu$ on $[0,\infty)$ equips the space of square integrable functions with inner product $\langle f,g\rangle_\mu = \int_0^\infty f(x)g(x)\mathrm{d}\mu(x)$, inducing a Hilbert space structure $\mathcal{H}_\mu$ and corresponding norm $\|f\|_{L_2(\mu)} = \langle f,f\rangle_\mu^{1/2}$.

**Polynomial Basis Expansion.**   Any $N$-dimensional subspace $\mathcal{G}$ of this function space is a suitable candidate for the approximation. The parameter $N$ corresponds to the order of the approximation, or the size of the compression; the projected history can be represented by the $N$ coefficients of its expansion in any basis of $\mathcal{G}$. For the remainder of this paper, we use the polynomials as a natural basis, so that $\mathcal{G}$ is the set of polynomials of degree less than $N$. We note that the polynomial basis is very general; for example, the Fourier basis $\sin(nx),\cos(nx)$ can be seen as polynomials on the unit circle $(e^{2\pi ix})^n$ (cf. Appendix D.4). In Appendix C, we additionally formalize a more general framework that allows different bases other than polynomials by tilting the measure with another function.

**Online Approximation.**   Since we care about approximating $f_{\leq t}$ for every time $t$, we also let the measure vary through time. For every $t$, let $\mu^{(t)}$ be a measure supported on $(-\infty,t]$ (since $f_{\leq t}$ is only defined up to time $t$). Overall, we seek some $g^{(t)} \in \mathcal{G}$ that minimizes $\|f_{\leq t} - g^{(t)}\|_{L_2(\mu^{(t)})}$. Intuitively, the measure $\mu$ controls the importance of various parts of the input domain, and the basis defines the allowable approximations. The challenge is how to solve the optimization problem in closed form given $\mu^{(t)}$, and how these coefficients can be maintained online as $t \to \infty$.

## 2.2   General HiPPO framework

We provide a brief overview of the main ideas behind solving this problem, which provides a surprisingly simple and general strategy for many measure families $\mu^{(t)}$. This framework builds upon a rich history of the well-studied *orthogonal polynomials* and related transforms in the signal processing literature. Our formal abstraction (Definition 1) departs from prior work on sliding transforms in several ways, which we discuss in detail in Appendix A.1. For example, our concept of the time-varying measure allows choosing $\mu^{(t)}$ more appropriately, which will lead to solutions with qualitatively different behavior. Appendix C contains the full details and formalisms of our framework.

**Calculating the projection through continuous dynamics.**   As mentioned, the approximated function can be represented by the $N$ coefficients of its expansion in any basis; the first key step is to choose a suitable basis $\{g_n\}_{n<N}$ of $\mathcal{G}$. Leveraging classic techniques from approximation theory, a natural basis is the set of orthogonal polynomials for the measure $\mu^{(t)}$, which forms an orthogonal basis of the subspace. Then the coefficients of the optimal basis expansion are simply $c_n^{(t)} := \langle f_{\leq t},g_n\rangle_{\mu^{(t)}}$.

The second key idea is to differentiate this projection in $t$, where differentiating through the integral (from the inner product $\langle f_{\leq t},g_n\rangle_{\mu^{(t)}}$) will often lead to a self-similar relation allowing $\frac{d}{dt}c_n(t)$ to be expressed in terms of $(c_k(t))_{k\in[N]}$ and $f(t)$. Thus the coefficients $c(t) \in \mathbb{R}^N$ should evolve as an ODE, with dynamics determined by $f(t)$.

**The HiPPO abstraction: online function approximation.**

**Definition 1.** *Given a time-varying measure family $\mu^{(t)}$ supported on $(-\infty,t]$, an $N$-dimensional subspace $\mathcal{G}$ of polynomials, and a continuous function $f : \mathbb{R}_{\geq 0} \to \mathbb{R}$, HiPPO defines a* projection *operator* $\mathrm{proj}_t$ *and a* coefficient extraction *operator* $\mathrm{coef}_t$ *at every time $t$, with the following properties:*

*(1)* $\mathrm{proj}_t$ *takes the function $f$ restricted up to time $t$, $f_{\leq t} := f(x)\,|_{x\leq t}$, and maps it to a polynomial $g^{(t)} \in \mathcal{G}$, that minimizes the approximation error $\|f_{\leq t} - g^{(t)}\|_{L_2(\mu^{(t)})}$.*

*(2)* $\mathrm{coef}_t : \mathcal{G} \to \mathbb{R}^N$ *maps the polynomial $g^{(t)}$ to the coefficients $c(t) \in \mathbb{R}^N$ of the basis of orthogonal polynomials defined with respect to the measure $\mu^{(t)}$.*

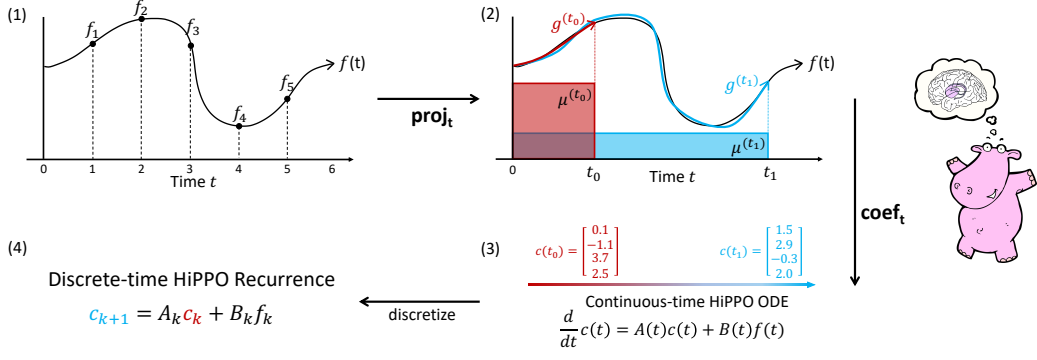

Figure 1: **Illustration of the HiPPO framework.** (1) For any function $f$, (2) at every time $t$ there is an optimal projection $g^{(t)}$ of $f$ onto the space of polynomials, with respect to a measure $\mu^{(t)}$ weighing the past. (3) For an appropriately chosen basis, the corresponding coefficients $c(t) \in \mathbb{R}^N$ representing a compression of the history of $f$ satisfy linear dynamics. (4) Discretizing the dynamics yields an efficient closed-form recurrence for online compression of time series $(f_k)_{k \in \mathbb{N}}$.

*The composition* $\mathrm{coef} \circ \mathrm{proj}$ *is called* hippo, *which is an operator mapping a function* $f : \mathbb{R}_{\geq 0} \to \mathbb{R}$ *to the optimal projection coefficients* $c : \mathbb{R}_{\geq 0} \to \mathbb{R}^N$, *i.e.* $(\mathrm{hippo}(f))(t) = \mathrm{coef}_t(\mathrm{proj}_t(f))$.

For each $t$, the problem of optimal projection $\mathrm{proj}_t(f)$ is well-defined by the above inner products, but this is intractable to compute naively. Our derivations (Appendix D) will show that the coefficient function $c(t) = \mathrm{coef}_t(\mathrm{proj}_t(f))$ has the form of an ODE satisfying $\frac{d}{dt}c(t) = A(t)c(t) + B(t)f(t)$ for some $A(t) \in \mathbb{R}^{N \times N}$, $B(t) \in \mathbb{R}^{N \times 1}$. Thus our results show how to tractably obtain $c^{(t)}$ *online* by solving an ODE, or more concretely by running a discrete recurrence. When discretized, HiPPO takes in a sequence of real values and produces a sequence of $N$-dimensional vectors.

Figure 1 illustrates the overall framework when we use uniform measures. Next, we give our main results showing hippo for several concrete instantiations of the framework.

## 2.3 High Order Projection: Measure Families and HiPPO ODEs

Our main theoretical results are instantiations of HiPPO for various measure families $\mu^{(t)}$. We provide two examples of natural sliding window measures and the corresponding projection operators. The unified perspective on memory mechanisms allows us to derive these closed-form solutions with the same strategy, provided in Appendices D.1,D.2. The first explains the core Legendre Memory Unit (LMU) [71] update in a principled way and characterizes its limitations, while the other is novel, demonstrating the generality of the HiPPO framework. Appendix D contrasts the tradeoffs of these measures (Fig. 5), contains proofs of their derivations, and derives additional HiPPO formulas for other bases such as Fourier (recovering the Fourier Recurrent Unit [79]) and Chebyshev.

The **translated Legendre (LegT)** measures assign uniform weight to the most recent history $[t-\theta,t]$. There is a hyperparameter $\theta$ representing the length of the sliding window, or the length of history that is being summarized. The **translated Laguerre (LagT)** measures instead use the exponentially decaying measure, assigning more importance to recent history.

$$\textbf{LegT}: \mu^{(t)}(x) = \frac{1}{\theta}\mathbb{I}_{[t-\theta,t]}(x) \qquad \textbf{LagT}: \mu^{(t)}(x) = e^{-(t-x)}\mathbb{I}_{(-\infty,t]}(x) = \begin{cases} e^{x-t} & \text{if } x \leq t \\ 0 & \text{if } x > t \end{cases}$$

**Theorem 1.** *For LegT and LagT, the* hippo *operators satisfying Definition 1 are given by linear time-invariant (LTI) ODEs* $\frac{d}{dt}c(t) = -Ac(t) + Bf(t)$, *where* $A \in \mathbb{R}^{N \times N}, B \in \mathbb{R}^{N \times 1}$:

**LegT**:
$$A_{nk} = \frac{1}{\theta}\begin{cases} (-1)^{n-k}(2n+1) & \text{if } n \geq k \\ 2n+1 & \text{if } n \leq k \end{cases}, \quad B_n = \frac{1}{\theta}(2n+1)(-1)^n \qquad \qquad \textbf{LagT}: \quad A_{nk} = \begin{cases} 1 & \text{if } n \geq k \\ 0 & \text{if } n < k \end{cases}, \quad B_n = 1 \quad (2)$$
$$(1)$$

Equation (1) proves the LMU update [71, equation (1)]. Additionally, our derivation (Appendix D.1) shows that outside of the projections, there is another source of approximation. This sliding window update rule requires access to $f(t-\theta)$, which is no longer available; it instead assumes that the current coefficients $c(t)$ are an accurate enough model of the function $f(x)_{x\leq t}$ that $f(t-\theta)$ can be recovered.

## 2.4 HiPPO recurrences: from Continuous to Discrete Time with ODE Discretization

Since actual data is inherently discrete (e.g. sequences and time series), we discuss how the HiPPO projection operators can be discretized using standard techniques, so that the continuous-time HiPPO ODEs become discrete-time linear recurrences.

In the continuous case, these operators consume an input function $f(t)$ and produce an output function $c(t)$. The discrete time case (i) consumes an input sequence $(f_k)_{k\in\mathbb{N}}$, (ii) implicitly defines a function $f(t)$ where $f(k \cdot \Delta t) = f_k$ for some step size $\Delta t$, (iii) produces a function $c(t)$ through the ODE dynamics, and (iv) discretizes back to an output sequence $c_k := c(k\cdot\Delta t)$.

The basic method of discretizating an ODE $\frac{d}{dt}c(t) = u(t, c(t), f(t))$ chooses a step size $\Delta t$ and performs the discrete updates $c(t + \Delta t) = c(t) + \Delta t \cdot u(t, c(t), f(t))$.[3] In general, this process is sensitive to the *discretization step size* hyperparameter $\Delta t$.

Finally, we note that this provides a way to seamlessly handle timestamped data, even with missing values: the difference between timestamps indicates the (adaptive) $\Delta t$ to use in discretization [13]. Appendix B.3 contains a full discussion of discretization.

## 2.5 Low Order Projection: Memory Mechanisms of Gated RNNs

As a special case, we consider what happens if we do not incorporate higher-order polynomials in the projection problem. Specifically, if $N = 1$, then the discretized version of HiPPO-LagT (2) becomes $c(t + \Delta t) = c(t) + \Delta t(-Ac(t) + Bf(t)) = (1 - \Delta t)c(t) + \Delta tf(t)$, since $A = B = 1$. If the inputs $f(t)$ can depend on the hidden state $c(t)$ and the discretization step size $\Delta t$ is chosen adaptively (as a function of input $f(t)$ and state $c(t)$), as in RNNs, then this becomes exactly a *gated* RNN. For instance, by stacking multiple units in parallel and choosing a specific update function, we obtain the GRU update cell as a special case.[4] In contrast to HiPPO which uses one hidden feature and projects it onto high order polynomials, these models use many hidden features but only project them with degree 1. This view sheds light on these classic techniques by showing how they can be derived from first principles.

## 3 HiPPO-LegS: Scaled Measures for Timescale Robustness

Exposing the tight connection between online function approximation and memory allows us to produce memory mechanisms with better theoretical properties, simply by choosing the measure appropriately. Although sliding windows are common in signal processing (Appendix A.1), a more intuitive approach for memory should *scale* the window over time to avoid forgetting.

Our novel **scaled Legendre measure (LegS)** assigns uniform weight to all history $[0,t]$: $\mu^{(t)} = \frac{1}{t}\mathbb{I}_{[0,t]}$. App D, Fig. 5 compares LegS, LegT, and LagT visually, showing the advantages of the scaled measure.

Simply by specifying the desired measure, specializing the HiPPO framework (Sections 2.2, 2.4) yields a new memory mechanism (proof in Appendix D.3).

**Theorem 2.** *The continuous-* (3) *and discrete-* (4) *time dynamics for **HiPPO-LegS** are:*

$$\frac{d}{dt}c(t) = -\frac{1}{t}Ac(t) + \frac{1}{t}Bf(t) \quad (3)$$

$$c_{k+1} = \left(1 - \frac{A}{k}\right)c_k + \frac{1}{k}Bf_k \quad (4)$$

$$A_{nk} = \begin{cases} (2n+1)^{1/2}(2k+1)^{1/2} & if\, n > k \\ n+1 & if\, n = k\,, \\ 0 & if\, n < k \end{cases} \qquad B_n = (2n+1)^{\frac{1}{2}}$$

We show that HiPPO-LegS enjoys favorable theoretical properties: it is invariant to input timescale, is fast to compute, and has bounded gradients and approximation error. All proofs are in Appendix E.

**Timescale robustness.** As the window size of LegS is adaptive, projection onto this measure is intuitively robust to timescales. Formally, the HiPPO-LegS operator is *timescale-equivariant*: dilating the input $f$ does not change the approximation coefficients.

**Proposition 3.** *For any scalar $\alpha > 0$, if $h(t) = f(\alpha t)$, then $\mathrm{hippo}(h)(t) = \mathrm{hippo}(f)(\alpha t)$. In other words, if $\gamma : t \mapsto \alpha t$ is any dilation function, then $\mathrm{hippo}(f \circ \gamma) = \mathrm{hippo}(f) \circ \gamma$.*

Informally, this is reflected by HiPPO-LegS having *no timescale hyperparameters*; in particular, the discrete recurrence (4) is invariant to the discretization step size.[5] By contrast, LegT has a hyperparameter $\theta$ for the window size, and both LegT and LagT have a step size hyperparameter $\Delta t$ in the discrete time case. This hyperparameter is important in practice; Section 2.5 showed that $\Delta t$ relates to the gates of RNNs, which are known to be sensitive to their parameterization [31, 39, 66]. We empirically demonstrate the benefits of timescale robustness in Section 4.2.

**Computational efficiency.** In order to compute a single step of the discrete HiPPO update, the main operation is multiplication by the (discretized) square matrix $A$. More general discretization specifically requires fast multiplication for any matrix of the form $I + \Delta t \cdot A$ and $(I - \Delta t \cdot A)^{-1}$ for arbitrary step sizes $\Delta t$. Although this is generically a $O(N^2)$ operation, LegS operators use a fixed $A$ matrix with special structure that turns out to have fast multiplication algorithms for any discretization.[6]

**Proposition 4.** *Under any generalized bilinear transform discretization (cf. Appendix B.3), each step of the HiPPO-LegS recurrence in equation (4) can be computed in $O(N)$ operations.*

Section 4.3 validates the efficiency of HiPPO layers in practice, where unrolling the discretized versions of Theorem 2 is 10x faster than standard matrix multiplication as done in standard RNNs.

**Gradient flow.** Much effort has been spent to alleviate the *vanishing gradient problem* in RNNs [56], where backpropagation-based learning is hindered by gradient magnitudes decaying exponentially in time. As LegS is designed for memory, it avoids the vanishing gradient issue.

**Proposition 5.** *For any times $t_0 < t_1$, the gradient norm of HiPPO-LegS operator for the output at time $t_1$ with respect to input at time $t_0$ is $\left\| \frac{\partial c(t_1)}{\partial f(t_0)} \right\| = \Theta(1/t_1)$.*

**Approximation error bounds.** The error rate of LegS decreases with the smoothness of the input.

**Proposition 6.** *Let $f : \mathbb{R}_+ \to \mathbb{R}$ be a differentiable function, and let $g^{(t)} = \mathrm{proj}_t(f)$ be its projection at time $t$ by HiPPO-LegS with maximum polynomial degree $N-1$. If $f$ is $L$-Lipschitz then $\left\| f_{\leq t} - g^{(t)} \right\| = O(tL/\sqrt{N})$. If $f$ has order-$k$ bounded derivatives then $\left\| f_{\leq t} - g^{(t)} \right\| = O(t^k N^{-k+1/2})$.*

## 4 Empirical Validation

The HiPPO dynamics are simple recurrences that can be easily incorporated into various models. We validate three claims that suggest that when incorporated into a simple RNN, these methods–especially HiPPO-LegS–yield a recurrent architecture with improved memory capability. In Section 4.1, the HiPPO-LegS RNN outperforms other RNN approaches in benchmark long-term dependency tasks for RNNs. Section 4.2 shows that HiPPO-LegS RNN is much more robust to timescale shifts compared to other RNN and neural ODE models. Section 4.3 validates the distinct theoretical advantages of the HiPPO-LegS memory mechanism, allowing fast and accurate online function reconstruction over millions of time steps. Experiment details and additional results are described in Appendix F.

**Model Architectures.** We first describe briefly how HiPPO memory updates can be incorporated into a simple neural network architecture, yielding a simple RNN model reminiscent of the classic LSTM. Given inputs $x_t$ or features thereof $f_t = u(x_t)$ in any model, the HiPPO framework can be used to memorize the history of features $f_t$. Thus, given any RNN update function $h_t = \tau(h_{t-1}, x_t)$, we simply replace $h_{t-1}$ with a projected version of the entire history of $h$, as described in Figure 2. The output of each cell is $h_t$, which can be passed through any downstream module (e.g. a classification head trained with cross-entropy) to produce predictions. We map the vector $h_{t-1}$ to 1D with a learned encoding before passing to $\mathrm{hippo}$ (full architecture in App. F.1).

## 4.1  Long-range Memory Benchmark Tasks

**Models and Baselines.**  We consider all of the HiPPO methods (**LegT**, **LagT**, and **LegS**). As we show that many different update dynamics seem to lead to LTI systems that give sensible results (Section 2.3), we additionally consider the **Rand** baseline that uses random $A$ and $B$ matrices (normalized appropriately) in its updates, to confirm that the precise derived dynamics are important. LegT additionally considers an additional hyperparameter $\theta$, which should be set to the timescale of the data if known a priori; to show the effect of the timescale, we set it to the ideal value as well as values that are too large and small. The **MGU** is a minimal gated architecture, equivalent to a GRU without the reset gate. The HiPPO architecture we use is simply the MGU with an additional hippo intermediate layer.

We also compare to several RNN baselines designed for long-term dependencies, including the **LSTM** [34], **GRU** [17], **expRNN** [48], and **LMU** [71].[7]

All methods have the same hidden size in our experiments. In particular, for simplicity and to reduce hyperparameters, HiPPO variants tie the memory size $N$ to the hidden state dimension $d$, so that all methods and baselines have a comparable number of hidden units and parameters. A more detailed comparison of model architectures is in Appendix F.1.

**Sequential Image Classification on Permuted MNIST.**  The permuted MNIST (pMNIST) task feeds inputs to a model pixel-by-pixel in the order of a fixed permutation. The model must process the entire image sequentially – with non-local structure – before outputting a classification label, requiring learning long-term dependencies.

Table 1 shows the validation accuracy on the pMNIST task for the instantiations of our framework and baselines. We highlight that LegS has the best performance of all models. While LegT is close at the optimal hyperparameter $\theta$, its performance can fall off drastically for a mis-specified window length. LagT also performs well at its best hyperparameter $\Delta t$.

Table 1 also compares test accuracy of our methods against reported results from the literature, where the LMU was the state-of-the-art for recurrent models. In addition to RNN-based baselines, other sequence models have been evaluated on this dataset, despite being against the spirit of the task because they have global receptive field instead of being strictly sequential. With a test accuracy of 98.3%, HiPPO-LegS sets a true state-of-the-art accuracy on the permuted MNIST dataset.

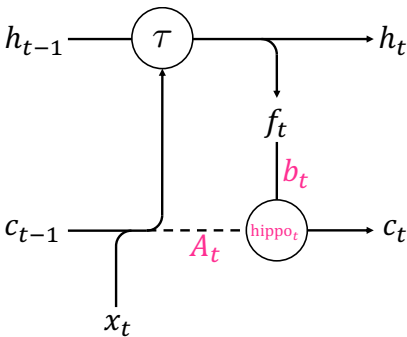

| Method | Val. acc. (%) |
|---|---|
| **-LegS** | **98.34** |
| -LagT | 98.15 |
| -LegT $\theta = 200$ | 98.0 |
| -LegT $\theta = 20$ | 91.75 |
| -Rand | 69.93 |
| LMU | 97.08 |
| ExpRNN | 94.67 |
| GRU | 93.04 |
| MGU | 89.37 |
| RNN | 52.98 |

| Model | Test acc. |
|---|---|
| **HiPPO-LegS** | **98.3** |
| LSTM [31] | 95.11 |
| r-LSTM [69] | 95.2 |
| Dilated RNN [10] | 96.1 |
| IndRNN [49] | 96.0 |
| URLSTM [31] | 96.96 |
| LMU [71] | 97.15 |
| Transformer [69] | 97.9 |
| TCN [5] | 97.2 |
| TrellisNet [6] | 98.13 |

Figure 2: HiPPO incorporated into a simple RNN model. hippo is the HiPPO memory operator which projects the history of the $f_t$ features depending on the chosen measure.

Table 1: (**Left**) pMNIST validation, average over 3 seeds. Top: Our methods. Bottom: RNN baselines. (**Right**) Reported test accuracies from previous works. Top: Our methods. Middle: Recurrent models. Bottom: Non-recurrent models requiring global receptive field.

**Copying task.**  This standard RNN task [3] directly tests memorization, where models must regurgitate a sequence of tokens seen at the beginning of the sequence. It is well-known that standard models such as LSTMs struggle to solve this task. Appendix F shows the loss for the Copying task with length $L = 200$. Our proposed update LegS solves the task almost perfectly, while LegT is very sensitive to the window length hyperparameter. As expected, most baselines make little progress.

## 4.2 Timescale Robustness of HiPPO-LegS

**Timescale priors.** Sequence models generally benefit from priors on the timescale, which take the form of additional hyperparameters in standard models. Examples include the "forget bias" of LSTMs which needs to be modified to address long-term dependencies [39, 66], or the discretization step size $\Delta t$ of HiPPO-Lag and HiPPO-LegT (Section 2.4). The experiments in Section 4.1 confirm their importance. Fig. 7 (Appendix) and Table 1 ablate these hyperparameters, showing that for example the sliding window length $\theta$ must be set correctly for LegT. Additional ablations for other hyperparameters are in Appendix F.

**Distribution shift in trajectory classification.** Recent trends in ML have stressed the importance of understanding robustness under distribution shift, when training and testing distributions are not i.i.d. For time series data, for example, models may be trained on EEG data from one hospital, but deployed at another using instruments with different sampling rates [62, 63]; or a time series may involve the same trajectory evolving at different speeds. Following Kidger et al. [40], we consider the Character Trajectories dataset [4], where the goal is to classify a character from a sequence of pen stroke measurements, collected from one user at a fixed sampling rate. To emulate timescale shift (e.g. testing on another user with slower handwriting), we consider two standard time series generation processes: (1) In the setting of sampling an underlying sequence at a fixed rate, we change the test sampling rate; crucially, the sequences are variable length so the models are unable to detect the sampling rate of the data. (2) In the setting of irregular-sampled (or missing) data with timestamps, we scale the test timestamps.

Recall that the HiPPO framework models the underlying data as a continuous function and interacts with discrete input only through the discretization. Thus, it seamlessly handles missing or irregularly-sampled data by simply evolving according to the given discretization step sizes (details in Appendix B.3). Combined with LegS timescale invariance (Prop. 3), we expect HiPPO-LegS to work automatically in all these settings. We note that the setting of missing data is a topic of independent interest and we compare against SOTA methods, including the GRU-D [11] which learns a decay between observations, and neural ODE methods which models segments between observations with an ODE.

Table 2 validates that standard models can go catastrophically wrong when tested on sequences at different timescales than expected. Though all methods achieve near-perfect accuracy ($\geq 95\%$) without distribution shift, aside from HiPPO-LegS, no method is able to generalize to unseen timescales.

Table 2: Test set accuracy on Character Trajectory classification on out-of-distribution timescales.

| Model | LSTM | GRU | GRU-D | ODE-RNN | NCDE | LMU | HiPPO-LegS |
|---|---|---|---|---|---|---|---|
| 100Hz $\rightarrow$ 200Hz | 31.9 | 25.4 | 23.1 | 41.8 | 44.7 | 6.0 | **88.8** |
| 200Hz $\rightarrow$ 100Hz | 28.2 | 64.6 | 25.5 | 31.5 | 11.3 | 13.1 | **90.1** |
| Missing values upsample | 24.4 | 28.2 | 5.5 | 4.3 | 63.9 | 39.3 | **94.5** |
| Missing values downsample | 34.9 | 27.3 | 7.7 | 7.7 | 69.7 | 67.8 | **94.9** |

## 4.3 Theoretical Validation and Scalability

We empirically show that HiPPO-LegS can scale to capture dependencies across millions of time steps, and its memory updates are computationally efficient (processing up to 470,000 time steps/s).

**Long-range function approximation.** We test the ability of different memory mechanisms in approximating an input function, as described in the problem setup in Section 2.1. The model only consists of the memory update (Section 3) and not the additional RNN architecture. We choose random samples from a continuous-time band-limited white noise process, with length $10^6$. The model is to traverse the input sequence, and then asked to reconstruct the input, while maintaining no more than 256 units in memory (Fig. 3). This is a difficult task; the LSTM fails with even sequences of length 1000 (MSE $\approx 0.25$). As shown in Table 3, both the LMU and HiPPO-LegS are able to accurately reconstruct the input function, validating that HiPPO can solve the function approximation problem even for very long sequences. Fig. 3 illustrates the function and its approximations, with HiPPO-LegS almost matching the input function while LSTM unable to do so.

**Speed.** HiPPO-LegS operator is computationally efficient both in theory (Section 3) and in practice. We implement the fast update in C++ with Pytorch binding and show in Table 3 that it can perform 470,000 time step updates per second on a single CPU core, 10x faster than the LSTM and LMU.[8]

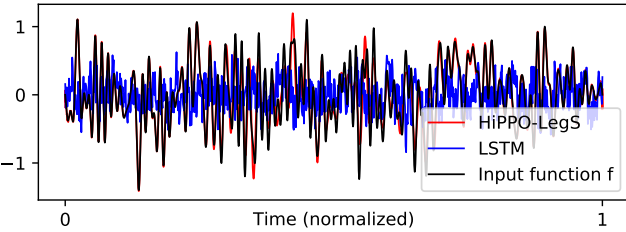

| Method | Error (MSE) | Speed (elements / sec) |
|---|---|---|
| LSTM | 0.25 | 35,000 |
| LMU | 0.05 | 41,000 |
| HiPPO-LegS | 0.02 | 470,000 |

Table 3: Function approximation error after 1 million time steps, with 256 hidden units.

Figure 3: Input function and its reconstructions.

## 4.4 Additional Experiments

We validate that the HiPPO memory updates also perform well on more generic sequence prediction tasks not exclusively focused on memory. Full results and details for these tasks are in Appendix F.

**Sentiment classification task on the IMDB movie review dataset.** Our RNNs with HiPPO memory updates perform on par with the LSTM, while other long-range memory approaches such as expRNN perform poorly on this more generic task (Appendix F.6).

**Mackey spin glass prediction.** This physical simulation task tests the ability to model chaotic dynamical systems. HiPPO-LegS outperforms the LSTM, LMU, and the best hybrid LSTM+LMU model from [71], reducing normalized MSE by $30\%$ (Appendix F.7).

## 5    Conclusion

We address the fundamental problem of memory in sequential data by proposing a framework (HiPPO) that poses the abstraction of optimal function approximation with respect to time-varying measures. In addition to unifying and explaining existing memory approaches, HiPPO unlocks a new method (HiPPO-LegS) that takes a first step toward timescale robustness and can efficiently handle dependencies across millions of time steps. We anticipate that the study of this core problem will be useful in improving a variety of sequence models, and are excited about future work on integrating our memory mechanisms with other models in addition to RNNs. We hope to realize the benefits of long-range memory on large-scale tasks such as speech recognition, video processing, and reinforcement learning.

## Broader Impact

Our work seeks to understand the foundation of memory in modeling sequential data, which may improve a wide range of applications, each with their own potential benefits and harms. For example, incorporating longer context in language modeling may improve the quality of automated customer services, helpdesks, and personal assistants, but might also facilitate spreading misinformation. Better video processing may produce more coherent video summary for visually impaired users, but might also make automatic surveillance easier.

Our framework presents a principled way to study memory of machine learning models. We speculate that this new representation could be a tool to study how potential biases in training data can get incorporated into the model. This in turn may give us a better handle on how to identify and potentially mitigate the effect of such biases in machine learning models. Though we currently do not have concrete ideas along this line, we encourage future investigation to better understand these learned representations to address fairness issues.

## Acknowledgments

We thank Avner May, Mayee Chen, Dan Fu, Aditya Grover, and Daniel Lévy for their helpful feedback. We gratefully acknowledge the support of DARPA under Nos. FA87501720095 (D3M), FA86501827865 (SDH), and FA86501827882 (ASED); NIH under No. U54EB020405 (Mobilize), NSF under Nos. CCF1763315 (Beyond Sparsity), CCF1563078 (Volume to Velocity), and 1937301 (RTML); ONR under No. N000141712266 (Unifying Weak Supervision); the Moore Foundation, NXP, Xilinx, LETI-CEA, Intel, IBM, Microsoft, NEC, Toshiba, TSMC, ARM, Hitachi, BASF, Accenture, Ericsson, Qualcomm, Analog Devices, the Okawa Foundation, American Family Insurance, Google Cloud, Stanford HAI AWS cloud credit, Swiss Re, and members of the Stanford DAWN project: Teradata, Facebook, Google, Ant Financial, NEC, VMWare, and Infosys. The U.S. Government is authorized to reproduce and distribute reprints for Governmental purposes notwithstanding any copyright notation thereon. Any opinions, findings, and conclusions or recommendations expressed in this material are those of the authors and do not necessarily reflect the views, policies, or endorsements, either expressed or implied, of DARPA, NIH, ONR, or the U.S. Government. Atri Rudra's research is supported by NSF grant CCF-1763481.

## Footnotes

[2]The LMU was originally motivated by spiking neural networks in modeling biological nervous systems; its derivation is not self-contained but a sketch can be pieced together from [71, 72, 73].

[3]This is known as the Euler method, used for illustration here; our experiments use the more numerically stable Bilinear and ZOH methods. Appendix B.3 provides a self-contained overview of our full discretization framework.

[4]The LSTM cell update is similar, with a parameterization known as "tied" gates [30].

[5](4) uses the Euler method for illustration; HiPPO-LegS is invariant to other discretizations (Appendix B.3).

[6]It is known that large families of structured matrices related to orthogonal polynomials are efficient [22].

[7]In our experiments, LMU refers to the architecture in [71] while LegT uses the one described in Fig. 2.

[8]The LMU is only known to be fast with the simple forward Euler discretization [71], but not with more sophisticated methods such as bilinear and ZOH that are required to reduce numerical errors for this task.

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
