[Supplementary Material]

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

# A   Related Work

Our work touches on a variety of topics and related work, which we explore in detail.

## A.1   Signal Processing and Orthogonal Polynomials

### A.1.1   Sliding transforms

The technical contributions in this work build on a rich history of approximation theory in signal processing. Our main framework – orthogonalizing functions with respect to time-varying measures (Section 2) – are related to "online" versions of classical signal processing transforms. In short, these methods compute *specific transforms* on *sliding windows* of *discrete sequences*. Concretely, they calculate $c_{n,k} = \sum_{i=0}^{N-1} f_{k+i} \psi(i,n)$ given signal $(f_k)$, where $\{\psi(i,n)\}$ is a discrete orthogonal transform. Our technical problem differs in several key aspects:

**Specific discrete transforms**   Examples of sliding transforms considered in the literature include the sliding DFT [26, 28, 36, 37], sliding DCT [43], sliding discrete (Walsh-)Hadamard transform [54, 55, 75], Haar [51], sliding discrete Hartley transform [44], and sliding discrete Chebyshev moments [12]. While each of these address a specific transform, we present a general approach (Section 2) that addresses several transforms at once. Furthermore, we are unaware of sliding transform algorithms for the OPs we consider here, in particular the Legendre and Laguerre polynomials. Our derivations in Appendix D cover Legendre, (generalized) Laguerre, Fourier, and Chebyshev continuous sliding transforms.

**Fixed-length sliding windows**   All mentioned works operate in the sliding window setting, where a fixed-size context window on the discrete signal is taken into account. Our measure-based abstraction for approximation allows considering a new type of *scaled* measure where the window size increases over time, leading to methods with qualitatively different theoretical (Section 3) and empirical properties (Section 4.2). We are not aware of any previous works addressing this scaled setting.

**Discrete vs. continuous time**   Even in the fixed-length sliding window case, our solutions to the "translated measure" problems (e.g., HiPPO-LegT Appendix D.1) solve a continuous-time sliding window problem on an underlying continuous signal, then discretize.

On the other hand, the sliding transform problems calculate transforms directly on a discrete stream. Discrete transforms are equivalent to calculating projection coefficients on a measure (equation (18)) by Gaussian quadrature, which assumes the discrete input is subsampled from a signal at the quadrature nodes [14]. However, since these nodes are non-uniformly spaced in general, the sliding discrete transform is not consistent with a discretization of an underlying continuous signal.

Thus, our main abstraction (Definition 1) has a fundamentally different interpretation than standard transforms, and our approach of first calculating the dynamics of the underlying continuous-time problem (e.g. equation (20)) is correspondingly new.

We remark that our novel *scaled measures* are fundamentally difficult to address with a standard discrete-time based approach. These discrete sliding methods require a fixed-size context in order to have consistent transform sizes, while the scaled measure would require solving transforms with an increasing number of input points over time.

### A.1.2   OPs in ML

More broadly, orthogonal polynomials and orthogonal polynomial transforms have recently found applications in various facets of machine learning. For example, Dao et al. [19] leverage the connection between orthogonal polynomials and quadrature to derive rules for computing kernel features in machine learning. More directly, [67] apply parametrized families of structured matrices directly inspired by orthogonal polynomial transforms ([22]) as layers in neural networks. Some particular families of orthogonal polynomials such as the Chebyshev polynomials have desirable approximation properties that find many well-known classical uses in numerical analysis and optimization. More recently, they have been applied to ML models such as graph convolutional neural networks[24], and generalizations such as Gegenbauer and Jacobi polynomials have been used to analyze optimization dynamics[7, 76]. Generalization of orthogonal polynomials and Fourier transform, expressed as products of butterfly

matrices, have found applications in automatic algorithm design [20], model compression [1], and replacing hand-crafted preprocessing in speech recognition [21]. Orthogonal polynomials are known to have various efficiency results [22], and we conjecture that Proposition 4 on the efficiency of HiPPO methods can be extended to arbitrary measures besides the ones considered in this work.

## A.2    Memory in Machine Learning

**Memory in sequence models**    Sequential or temporal data in areas such as language, reinforcement learning, and continual learning can involve increasingly long dependencies. However, direct parametric modeling cannot handle inputs of unknown and potentially unbounded lengths. Many modern solutions such as attention [70] and dilated convolutions [5], are functions on finite windows, thus sidestepping the need for an explicit memory representation. While this suffices for certain tasks, these approaches can only process a finite context window instead of an entire sequence. Naively increasing the window length poses significant compute and memory challenges. This has spurred various approaches to extend this fixed context window subjected to compute and storage constraints [6, 15, 18, 42, 59, 60, 64, 74].

We instead focus on the core problem of online processing and memorization of continuous and discrete signals, and anticipate that the study of this foundational problem will be useful in improving a variety of models.

**Recurrent memory**    Recurrent neural networks are a natural tool for modeling sequential data online, with the appealing property of having unbounded context; in other words they can summarize history indefinitely. However, due to difficulties in the optimization process (vanishing/exploding gradients [56]), particular care must be paid to endow them with longer memory. The ubiquitous LSTM [34] and simplifications such as the GRU [17] control the update with gates to smooth the optimization process. With more careful parametrization, the addition of gates alone make RNNs significantly more robust and able to address long-term dependencies [31]. Tallec and Ollivier [66] show that gates are in fact fundamental for recurrent dynamics by allowing time dilations. Many other approaches to endowing RNNs with better memory exist, such as noise injection [32] or non-saturating gates [9], which can suffer from instability issues. A long line of work controls the spectrum of the recurrent updates with (nearly-) orthogonal matrices to control gradients [3], but have been found to be less robust across different tasks [33].

## A.3    Directly related methods

**LMU**    The main result of the Legendre Memory Unit [71, 72, 73] is a direct instantiation of our framework using the LegT measure (Section 2.3). The original LMU is motivated by neurobiological advances and approaches the problem from the opposite direction as us: it considers approximating spiking neurons in the frequency domain, while we directly solve an interpretable optimization problem in the time domain. More specifically, they consider time-lagged linear time invariant (LTI) dynamical systems and approximate the dynamics with Padé approximants; Voelker et al. [71] observes that the result also has an interpretation in terms of Legendre polynomials, but not that it is the optimal solution to a natural projection problem. This approach involves heavier machinery, and we were not able to find a complete proof of the update mechanism [71, 72, 73].

In contrast, our approach directly poses the relevant online signal approximation problem, which ties to orthogonal polynomial families and leads to simple derivations of several related memory mechanisms (Appendix D). Our interpretation in time rather than frequency space, and associated derivation (Appendix D.1) for the LegT measure, reveals a different set of approximations stemming from the sliding window, which is confirmed empirically (Appendix F.8).

As the motivations of our work are substantially different from Voelker et al. [71], yet finds the same memory mechanism in a special case, we highlight the potential connection between these sequence models and biological nervous systems as an area of exploration for future work, such as alternative interpretations of our methods in the frequency domain.

We remark that the term LMU in fact refers to a specific recurrent neural network architecture, which interleaves the projection operator with other specific neural network components. By contrast, we use HiPPO to refer to the projection operator in isolation (Theorem 1), which is a function-to-function or sequence-to-sequence operator independent of model. HiPPO is integrated into an RNN architecture

in Section 4, with slight improvements to the LMU architecture, as ablated in Appendices F.2 and F.3. As a standalone module, HiPPO can be used as a layer in other types of models.

**Fourier Recurrent Unit** The Fourier Recurrent Unit (FRU) [79] uses Fourier basis (cosine and sine) to express the input signal, motivated by the discrete Fourier transform. In particular, each recurrent unit computes the discrete Fourier transform of the input signal for a randomly chosen frequency. It is not clear how discrete transform with respect to other bases (e.g., Legendre, Laguerre, Chebyshev) can in turn yield similar memory mechanisms. We show that FRU is also an instantiation of the HiPPO framework (Appendix D.4), where the Fourier basis can be viewed as orthogonal polynomials $z^n$ on the unit circle $\{z \colon |z| = 1\}$.

Zhang et al. [79] prove that if a timescale hyperparameter is chosen appropriately, FRU has bounded gradients, thus avoiding vanishing and exploding gradients. This essentially follows from the fact that $(1 - \Delta t)^T = \Theta(1)$ if the discretization step size $\Delta t = \Theta(\frac{1}{T})$ is chosen, if the time horizon $T$ is known (cf. Appendices B.3 and E). It is easily shown that this property is not intrinsic to the FRU but to sliding window methods, and is shared by all of our translated measure HiPPO methods (all but HiPPO-LegS in Appendix D). We show the stronger property that HiPPO-LegS, which uses scaling rather than sliding windows, also enjoys bounded gradient guarantees, without needing a well-specified timescale hyperparameter (Proposition 5).

**Neural ODEs** HiPPO produces linear ODEs that describe the dynamics of the coefficients. Recent work has also incorporated ODEs into machine learning models. Chen et al. [13] introduce neural ODEs, employing general nonlinear ODEs parameterized by neural networks in the context of normalizing flows and time series modeling. Neural ODEs have shown promising results in modeling irregularly sampled time series [40], especially when combined with RNNs [61]. Though neural ODEs are expressive [27, 78], due to their complex parameterization, they often suffer from slow training [29, 53, 58] because of their need for more complicated ODE solvers. On the other hand, HiPPO ODEs are linear and are fast to solve with classical discretization techniques in linear systems, such as Euler method, Bilinear method, and Zero-Order Hold (ZOH) [35].

## B  Technical Preliminaries

We collect here some technical background that will be used in presenting the general HiPPO framework and in deriving specific HiPPO update rules.

### B.1  Orthogonal Polynomials

Orthogonal polynomials are a standard tool for working with function spaces [14, 65]. Every measure $\mu$ induces a unique (up to a scalar) sequence of *orthogonal polynomials* (OPs) $P_0(x), P_1(x), \dots$ satisfying $\deg(P_i) = i$ and $\langle P_i, P_j \rangle_\mu := \int P_i(x) P_j(x) \mathrm{d}\mu(x) = 0$ for all $i \neq j$. This is the sequence found by orthogonalizing the monomial basis $\{x^i\}$ with Gram-Schmidt with respect to $\langle \cdot, \cdot \rangle_\mu$. The fact that OPs form an orthogonal basis is useful because the optimal polynomial $g$ of degree $\deg(g) < N$ that approximates a function $f$ is then given by

$$\sum_{i=0}^{N-1} c_i P_i(x) / \|P_i\|_\mu^2 \qquad \text{where } c_i = \langle f, P_i \rangle_\mu = \int f(x) P_i(x) \mathrm{d}\mu(x).$$

Classical OPs families comprise Jacobi (which include Legendre and Chebyshev polynomials as special cases), Laguerre, and Hermite polynomials. The Fourier basis can also be interpreted as OPs on the unit circle in the complex plane.

### B.1.1  Properties of Legendre Polynomials

**Legendre polynomials** Under the usual definition of the canonical Legendre polynomial $P_n$, they are orthogonal with respect to the measure $\omega^{\mathrm{leg}} = \mathbf{1}_{[-1,1]}$:

$$\frac{2n+1}{2} \int_{-1}^{1} P_n(x) P_m(x) \mathrm{d}x = \delta_{nm} \tag{5}$$

Also, they satisfy

$$P_n(1) = 1$$
$$P_n(-1) = (-1)^n.$$

**Shifted and Scaled Legendre polynomials**  We will also consider scaling the Legendre polynomials to be orthogonal on the interval $[0,t]$. A change of variables on (5) yields

$$(2n+1)\int_0^t P_n\left(\frac{2x}{t}-1\right)P_m\left(\frac{2x}{t}-1\right)\frac{1}{t}\mathrm{d}x = (2n+1)\int P_n\left(\frac{2x}{t}-1\right)P_m\left(\frac{2x}{t}-1\right)\omega^{\mathrm{leg}}\left(\frac{2x}{t}-1\right)\frac{1}{t}\mathrm{d}x$$

$$= \frac{2n+1}{2}\int P_n(x)P_m(x)\omega^{\mathrm{leg}}(x)\mathrm{d}x$$

$$= \delta_{nm}.$$

Therefore, with respect to the measure $\omega_t = \mathbf{1}_{[0,t]}/t$ (which is a probability measure for all $t$), the normalized orthogonal polynomials are

$$(2n+1)^{1/2}P_n\left(\frac{2x}{t}-1\right).$$

Similarly, the basis

$$(2n+1)^{1/2}P_n\left(2\frac{x-t}{\theta}+1\right)$$

is orthonormal for the uniform measure $\frac{1}{\theta}\mathbb{I}_{[t-\theta,t]}$.

In general, the orthonormal basis for any uniform measure consists of $(2n+1)^{\frac{1}{2}}$ times the corresponding linearly shifted version of $P_n$.

**Derivatives of Legendre polynomials**  We note the following recurrence relations on Legendre polynomials ([2, Chapter 12]):

$$(2n+1)P_n = P'_{n+1} - P'_{n-1}$$
$$P'_{n+1} = (n+1)P_n + xP'_n$$

The first equation yields

$$P'_{n+1} = (2n+1)P_n + (2n-3)P_{n-2} + ..., \tag{6}$$

where the sum stops at $P_0$ or $P_1$.

These equations directly imply

$$P'_n = (2n-1)P_{n-1} + (2n-5)P_{n-3} + ... \tag{7}$$

and

$$(x+1)P'_n(x) = P'_{n+1} + P'_n - (n+1)P_n$$
$$= nP_n + (2n-1)P_{n-1} + (2n-3)P_{n-2} + .... \tag{8}$$

These will be used in the derivations of the HiPPO-LegT and HiPPO-LegS updates, respectively.

### B.1.2   Properties of Laguerre Polynomials

The standard Laguerre polynomials $L_n(x)$ are defined to be orthogonal with respect to the weight function $e^{-x}$ supported on $[0,\infty)$, while the generalized Laguerre polynomials (also called associated Laguerre polynomials) $L_n^{(\alpha)}$ are defined to be orthogonal with respect to the weight function $x^\alpha e^{-x}$ also supported on $[0,\infty)$:

$$\int_0^\infty x^\alpha e^{-x} L_n^{(\alpha)}(x) L_m^{(\alpha)}(x)\mathrm{d}x = \frac{(n+\alpha)!}{n!}\delta_{n,m}. \tag{9}$$

Also, they satisfy

$$L_n^{(\alpha)}(0) = \binom{n+\alpha}{n} = \frac{\Gamma(n+\alpha+1)}{\Gamma(n+1)\Gamma(\alpha+1)}. \tag{10}$$

The standard Laguerre polynomials correspond to the case of $\alpha = 0$ of generalized Laguerre polynomials.

**Derivatives of generalized Laguerre polynomials**  We note the following recurrence relations on generalized Laguerre polynomials ([2, Chapter 13.2]):

$$\frac{\mathrm{d}}{\mathrm{d}x}L_n^{(\alpha)}(x) = -L_{n-1}^{(\alpha+1)}(x)$$

$$L_n^{(\alpha+1)}(x) = \sum_{i=0}^{n} L_i^{(\alpha)}(x).$$

These equations imply

$$\frac{d}{dt}L_n^{(\alpha)}(x) = -L_0^{(\alpha)}(x) - L_1^{(\alpha)}(x) - \cdots - L_{n-1}^{(\alpha)}(x).$$

### B.1.3   Properties of Chebyshev polynomials

Let $T_n$ be the classical Chebyshev polynomials (of the first kind), defined to be orthogonal with respect to the weight function $(1-x^2)^{1/2}$ supported on $[-1,1]$, and let $p_n$ be the normalized version of $T_n$ (i.e, with norm 1):

$$\omega^{\mathrm{cheb}} = (1-x^2)^{-1/2}\mathbb{I}_{(-1,1)},$$

$$p_n(x) = \sqrt{\frac{2}{\pi}}T_n(x) \qquad \text{for } n \geq 1,$$

$$p_0(x) = \frac{1}{\sqrt{\pi}}.$$

Note that $\omega^{\mathrm{cheb}}$ is not normalized (it integrates to $\pi$).

**Derivatives of Chebyshev polynomials**  The chebyshev polynomials satisfy

$$2T_n(x) = \frac{1}{n+1}\frac{d}{dx}T_{n+1}(x) - \frac{1}{n-1}\frac{d}{dx}T_{n-1}(x) \qquad n = 2,3,\dots.$$

By telescoping this series, we obtain

$$\frac{1}{n}T_n' = \begin{cases} 2(T_{n-1}+T_{n-3}+\cdots+T_2)+T_0 & n \text{ odd} \\ 2(T_{n-1}+T_{n-3}+\cdots+T_1) & n \text{ even} \end{cases}. \tag{11}$$

**Translated Chebyshev polynomials**  We will also consider shifting and scaling the Chebyshev polynomials to be orthogonal on the interval $[t-\theta,t]$ for fixed length $\theta$.

The normalized (probability) measure is

$$\omega(t,x) = \frac{2}{\theta\pi}\omega^{\mathrm{cheb}}\left(\frac{2(x-t)}{\theta}+1\right) = \frac{1}{\theta\pi}\left(\frac{x-t}{\theta}+1\right)^{-1/2}\left(-\frac{x-t}{\theta}\right)^{-1/2}\mathbb{I}_{(t-\theta,t)}.$$

The orthonormal polynomial basis is

$$p_n(t,x) = \sqrt{\pi}p_n\left(\frac{2(x-t)}{\theta}+1\right).$$

In terms of the original Chebyshev polynomials, these are

$$p_n(t,x) = \sqrt{2}T_n\left(\frac{2(x-t)}{\theta}+1\right) \qquad \text{for } n \geq 1,$$

$$p_0^{(t)} = T_0\left(\frac{2(x-t)}{\theta}+1\right).$$

### B.2   Leibniz Integral Rule

As part of our standard strategy for deriving HiPPO update rules (Appendix C), we will differentiate through integrals with changing limits. For example, we may wish to differentiate with respect to $t$ the expression $\int f(t,x)\mu(t,x)\mathrm{d}x = \int_0^t f(t,x)\frac{1}{t}\mathrm{d}x$ when analyzing the scaled Legendre (LegS) measure.

Differentiating through such integrals can be formalized by the Leibniz integral rule, the basic version of which states that

$$\frac{\partial}{\partial t}\int_{\alpha(t)}^{\beta(t)} f(x,t)\mathrm{d}x = \int_{\alpha(t)}^{\beta(t)}\frac{\partial}{\partial t}f(x,t)\mathrm{d}x - \alpha'(t)f(\alpha(t),t) + \beta'(t)f(\beta(t),t).$$

We elide over the formalisms in our derivations (Appendix D) and instead use the following trick. We replace integrand limits with an indicator function; and using the Dirac delta function $\delta$ when differentiating (i.e., using the formalism of distributional derivatives). For example, the above formula can be derived succinctly with this trick:

$$\frac{\partial}{\partial t}\int_{\alpha(t)}^{\beta(t)} f(x,t)\mathrm{d}x = \frac{\partial}{\partial t}\int f(x,t)\mathbb{I}_{[\alpha(t),\beta(t)]}(x)\mathrm{d}x$$

$$= \int\frac{\partial}{\partial t}f(x,t)\mathbb{I}_{[\alpha(t),\beta(t)]}(x)\mathrm{d}x + \int f(x,t)\frac{\partial}{\partial t}\mathbb{I}_{[\alpha(t),\beta(t)]}(x)\mathrm{d}x$$

$$= \int\frac{\partial}{\partial t}f(x,t)\mathbb{I}_{[\alpha(t),\beta(t)]}(x)\mathrm{d}x + \int f(x,t)(\beta'(t)\delta_{\beta(t)}(x) - \alpha'(t)\delta_{\alpha(t)})(x)\mathrm{d}x$$

$$= \int_{\alpha(t)}^{\beta(t)}\frac{\partial}{\partial t}f(x,t)\mathrm{d}x - \alpha'(t)f(\alpha(t),t) + \beta'(t)f(\beta(t),t).$$

### B.3  ODE Discretization

In our framework, time series inputs will be modeled with a continuous function and then discretized. Here we provide some background on ODE discretization methods, including a new discretization that applies to a specific type of ODE that our new method encounters.

The general formulation of an ODE is $\frac{d}{dt}c(t) = f(t,c(t))$. We will also focus on the linear time-invariant ODE of the form $\frac{d}{dt}c(t) = Ac(t) + Bf(t)$ for some input function $f(t)$, as a special case. The general methodology for discretizing the ODE, for step size $\Delta t$, is to rewrite the ODE as

$$c(t+\Delta t) - c(t) = \int_t^{t+\Delta t} f(s,c(s))\mathrm{d}s, \tag{12}$$

then approximate the RHS integral.

Many ODE discretization methods corresponds to different ways to approximate the RHS integral:

**Euler (aka forward Euler).**  To approximate the RHS of equation (12), keep the left endpoint $\Delta t f(t,c(t))$. For the linear ODE, we get:

$$c(t+\Delta t) = (I+\Delta tA)c(t) + \Delta tBf(t).$$

**Backward Euler.**  To approximate the RHS of equation (12), keep the right endpoint $\Delta t f(t+\Delta t,c(t+\Delta t))$. For the linear ODE, we get the linear equation and the update:

$$c(t+\Delta t) - \Delta tAc(t+\Delta t) = c(t) + \Delta tBf(t)$$
$$c(t+\Delta t) = (I-\Delta tA)^{-1}c(t) + \Delta t(I-\Delta tA)^{-1}Bf(t).$$

**Bilinear (aka Trapezoid rule, aka Tustin's method).**  To approximate the RHS of equation (12), average the endpoints $\Delta t\frac{f(t,c(t))+f(t+\Delta t,c(t+\Delta t))}{2}$. For the linear ODE, again we get a linear equation and the update:

$$c(t+\Delta t) - \frac{\Delta t}{2}Ac(t+\Delta t) = (I+\Delta t/2A)c(t) + \Delta tBf(t)$$
$$c(t+\Delta t) = (I-\Delta t/2A)^{-1}(I+\Delta t/2A)c(t) + \Delta t(I-\Delta t/2A)^{-1}Bf(t).$$

**Generalized Bilinear Transformation (GBT).**   This method [77] approximates the RHS of equation (12) by taking a weighted average of the endpoints $\Delta t[(1-\alpha)f(t,c(t))+\alpha f(t+\Delta t,c(t+\Delta t))]$, for some parameter $\alpha \in [0,1]$. For the linear ODE, again we get a linear equation and the update:

$$c(t+\Delta t)-\Delta t\alpha Ac(t+\Delta t)=(I+\Delta t(1-\alpha)A)c(t)+\Delta tBf(t)$$

$$c(t+\Delta t)=(I-\Delta t\alpha A)^{-1}(I+\Delta t(1-\alpha)A)c(t)+\Delta t(I-\Delta t\alpha A)^{-1}Bf(t). \quad (13)$$

GBT generalizes the three methods mentioned above: forward Euler corresponds to $\alpha=0$, backward Euler to $\alpha=1$, and bilinear to $\alpha=1/2$.

We also note another method called *Zero-order Hold* (ZOH) [23] that specializes to linear ODEs. The RHS of equation (12) is calculated in closed-form assuming constant input $f$ between $t$ and $t+\Delta t$. This yields the update $c(t+\Delta t)=e^{\Delta tA}c(t)+\left(\int_{\tau=0}^{\Delta t}e^{\tau A}\mathrm{d}\tau\right)Bf(t)$. If $A$ is invertible, this can be simplified as $c(t+\Delta t)=e^{\Delta tA}c(t)+A^{-1}(e^{\Delta tA}-I)Bf(t)$.

**HiPPO-LegS invariance to discretization step size.**   In the case of HiPPO-LegS, we have a linear ODE of the form $\frac{d}{dt}c(t)=\frac{1}{t}Ac(t)+\frac{1}{t}Bf(t)$. Adapting the GBT discretization (which generalizes forward/backward Euler and bilinear) to this linear ODE, we obtain:

$$c(t+\Delta t)-\Delta t\alpha\frac{1}{t+\Delta t}Ac(t+\Delta t)=\left(I+\Delta t(1-\alpha)\frac{1}{t}A\right)c(t)+\Delta t\frac{1}{t}Bf(t)$$

$$c(t+\Delta t)=\left(I-\frac{\Delta t}{t+\Delta t}\alpha A\right)^{-1}\left(I+\frac{\Delta t}{t}(1-\alpha)A\right)c(t)+\frac{\Delta t}{t}\left(I-\frac{\Delta t}{t+\Delta t}\alpha A\right)^{-1}Bf(t).$$

We highlight that this system is invariant to the discretization step size $\Delta t$. Indeed, if $c^{(k)}:=c(k\Delta t)$ and $f_k:=f(k\Delta t)$ then we have the recurrence

$$c^{(k+1)}=\left(I-\frac{1}{k+1}\alpha A\right)^{-1}\left(I+\frac{1}{k}(1-\alpha)A\right)c^{(k)}+\frac{1}{k}\left(I-\frac{1}{k+1}\alpha A\right)^{-1}Bf_k,$$

which does not depend on $\Delta t$.

**Ablation: comparison between different discretization methods**   To understand the impact of approximation error in discretization, in Fig. 4, we show the absolute error for the HiPPO-LegS updates in function approximation (Appendix F.8) for different discretization methods: forward Euler, backward Euler, and bilinear. The bilinear method generally provide sufficiently accurate approximation. We will use bilinear as the discretization method for the LegS updates for the experiments.

Figure 4: Absolute error for different discretization methods. Forward and backward Euler are generally not very accurate, while bilinear yields more accurate approximation.

## C   General HiPPO Framework

We present the general HiPPO framework, as described in Section 2, in more details. We also generalize it to include bases other than polynomials.

Given a time-varying measure family $\mu^{(t)}$ supported on $(-\infty, t]$, a sequence of basis functions $\mathcal{G} = \text{span}\{g_n^{(t)}\}_{n \in [N]}$, and a continuous function $f : \mathbb{R}_{\geq 0} \to \mathbb{R}$, HiPPO defines an operator that maps $f$ to the optimal projection coefficients $c : \mathbb{R}_{\geq 0} \to \mathbb{R}^N$, such that

$$g^{(t)} := \text{argmin}_{g \in \mathcal{G}} \|f_{\leq t} - g\|_{\mu^{(t)}}, \qquad \text{and} \qquad g^{(t)} = \sum_{n=0}^{N-1} c_n(t) g_n^{(t)}.$$

The first step refers to the $\text{proj}_t$ operator and the second the $\text{coef}_t$ operator in Definition 1.

We focus on the case where the coefficients $c(t)$ has the form of a linear ODE satisfying $\frac{d}{dt} c(t) = A(t) c(t) + B(t) f(t)$ for some $A(t) \in \mathbb{R}^{N \times N}, B(t) \in \mathbb{R}^{N \times 1}$.

We first describe the parameters of the $\text{hippo}$ operator (a measure and basis) in more detail in Appendix C.1. We define the projection $\text{proj}_t$ and coefficient $\text{coef}_t$ operators in Appendix C.2. Then we give a general strategy to calculate these coefficients $c(t)$, by deriving a differential equation that governs the coefficient dynamics (Appendix C.3). Finally we discuss how to turn the continuous $\text{hippo}$ operator into a discrete one that can be applied to sequence data (Appendix C.4).

## C.1 Measure and Basis

We describe and motivate the ingredients of HiPPO in more detail here. Recall that the high level goal is online function approximation; this requires both a set of valid approximations and a notion of approximation quality.

**Approximation Measures** At every $t$, the approximation quality is defined with respect to a measure $\mu^{(t)}$ supported on $(-\infty, t]$. We seek some polynomial $g^{(t)}$ of degree at most $N - 1$ that minimizes the error $\|f_{x \leq t} - g^{(t)}\|_{L_2(\mu^{(t)})}$. Intuitively, this measure $\mu^{(t)}$ governs how much to weigh every time in the past. For simplicity, we assume that the measures $\mu^{(t)}$ are sufficiently smooth across their domain as well as in time; in particular, they have densities $\omega(t, x) := \frac{d\mu^{(t)}}{d\lambda}(x)$ with respect to the Lebesgue measure $d\lambda(x) := dx$ such that $\omega$ is $C^1$ almost everywhere. Thus integrating against $d\mu^{(t)}(x)$ can be rewritten as integrating against $\omega(t, x) dx$.

We also assume for simplicity that the measures $\mu^{(t)}$ are normalized to be probability measures; arbitrary scaling does not affect the optimal projection.

**Orthogonal polynomial basis** Let $\{P_n\}_{n \in \mathbb{N}}$ denote a sequence of orthogonal polynomials with respect to some base measure $\mu$. Similarly define $\{P_n^{(t)}\}_{n \in \mathbb{N}}$ to be a sequence of orthogonal polynomials with respect to the time-varying measure $\mu^{(t)}$. Let $p_n^{(t)}$ be the normalized version of $P_n^{(t)}$ (i.e., have norm 1), and define

$$p_n(t, x) = p_n^{(t)}(x). \tag{14}$$

Note that the $P_n^{(t)}$ are not required to be normalized, while the $p_n^{(t)}$ are.

**Tilted measure and basis** Our goal is simply to store a compressed representation of functions, which can use any basis, not necessarily OPs. For any scaling function

$$\chi(t, x) = \chi^{(t)}(x), \tag{15}$$

the functions $p_n(x)\chi(x)$ are orthogonal with respect to the density $\omega/\chi^2$ at every time $t$. Thus, we can choose this alternative basis and measure to perform the projections.

To formalize this tilting with $\chi$, define $\nu^{(t)}$ to be the normalized measure with density proportional to $\omega^{(t)}/(\chi^{(t)})^2$. We will calculate the normalized measure and the orthonormal basis for it. Let

$$\zeta(t) = \int \frac{\omega}{\chi^2} = \int \frac{\omega^{(t)}(x)}{(\chi^{(t)}(x))^2} dx \tag{16}$$

be the normalization constant, so that $\nu^{(t)}$ has density $\frac{\omega^{(t)}}{\zeta(t)(\chi^{(t)})^2}$. If $\chi(t, x) = 1$ (no tilting), this constant is $\zeta(t) = 1$. In general, we assume that $\zeta$ is constant for all $t$; if not, it can be folded into $\chi$ directly.

Next, note that (dropping the dependence on $x$ inside the integral for shorthand)

$$
\begin{aligned}
\left\| \zeta(t)^{\frac{1}{2}} p_n^{(t)} \chi^{(t)} \right\|_{\nu^{(t)}}^2 &= \int \left( \zeta(t)^{\frac{1}{2}} p_n^{(t)} \chi^{(t)} \right)^2 \frac{\omega^{(t)}}{\zeta(t)(\chi^{(t)})^2} \\
&= \int (p_n^{(t)})^2 \omega^{(t)} \\
&= \left\| p_n^{(t)} \right\|_{\mu^{(t)}}^2 = 1.
\end{aligned}
$$

Thus we define the orthogonal basis for $\nu^{(t)}$

$$
g_n^{(t)} = \lambda_n \zeta(t)^{\frac{1}{2}} p_n^{(t)} \chi^{(t)}, \quad n \in \mathbb{N}. \tag{17}
$$

We let each element of the basis be scaled by a $\lambda_n$ scalar, for reasons discussed soon, since arbitrary scaling does not change orthogonality:

$$
\langle g_n^{(t)}, g_m^{(t)} \rangle_{\nu^{(t)}} = \lambda_n^2 \delta_{n,m}
$$

Note that when $\lambda_n = \pm 1$, the basis $\{g_n^{(t)}\}$ is an orthonormal basis with respect to the measure $\nu^{(t)}$, at every time $t$. Notationally, let $g_n(t,x) := g_n^{(t)}(x)$ as usual.

We will only use this tilting in the case of Laguerre (Appendix D.2 and Chebyshev (Appendix D.5).

Note that in the case $\chi = 1$ (i.e., no tilting), we also have $\zeta = 1$ and $g_n = \lambda_n p_n$ (for all $t,x$).

## C.2   The Projection and Coefficients

Given a choice of measures and basis functions, we next see how the coefficients $c(t)$ can be computed.

**Input: Function**   We are given a $C^1$-smooth function $f : [0,\infty) \to \mathbb{R}$ which is seen *online*, for which we wish to maintain a compressed representation of its history $f(x)_{\leq t} = f(x)_{x \leq t}$ at every time $t$.

**Output: Approximation Coefficients**   The function $f$ can be approximated by storing its coefficients with respect to the basis $\{g_n\}_{n<N}$. For example, in the case of no tilting $\chi = 1$, this encodes the optimal polynomial approximation of $f$ of degree less than $N$. In particular, at time $t$ we wish to represent $f_{\leq t}$ as a linear combination of polynomials $g_n^{(t)}$. Since the $g_n^{(t)}$ are orthogonal with respect to the Hilbert space defined by $\langle \cdot, \cdot \rangle_{\nu^{(t)}}$, it suffices to calculate coefficients

$$
\begin{aligned}
c_n(t) &= \langle f_{\leq t}, g_n^{(t)} \rangle_{\nu^{(t)}} \\
&= \int f g_n^{(t)} \frac{\omega^{(t)}}{\zeta(t)(\chi^{(t)})^2} \\
&= \zeta(t)^{-\frac{1}{2}} \lambda_n \int f p_n^{(t)} \frac{\omega^{(t)}}{\chi^{(t)}}.
\end{aligned} \tag{18}
$$

**Reconstruction**   At any time $t$, $f_{\leq t}$ can be explicitly reconstructed as

$$
\begin{aligned}
f_{\leq t} \approx g^{(t)} &:= \sum_{n=0}^{N-1} \langle f_{\leq t}, g_n^{(t)} \rangle_{\nu^{(t)}} \frac{g_n^{(t)}}{\|g_n^{(t)}\|_{\nu^{(t)}}^2} \\
&= \sum_{n=0}^{N-1} \lambda_n^{-2} c_n(t) g_n^{(t)} \\
&= \sum_{n=0}^{N-1} \lambda_n^{-1} \zeta^{\frac{1}{2}} c_n(t) p_n^{(t)} \chi^{(t)}.
\end{aligned} \tag{19}
$$

Equation (19) is the $\text{proj}_t$ operator; given the measure and basis parameters, it defines the optimal approximation of $f_{\leq t}$.

The $\text{coef}_t$ operator simply extracts the vector of coefficients $c(t) = (c_n(t))_{n \in [N]}$.

### C.3 Coefficient Dynamics: the hippo Operator

For the purposes of end-to-end models consuming an input function $f(t)$, the coefficients $c(t)$ are enough to encode information about the history of $f$ and allow online predictions. Therefore, defining $c(t)$ to be the vector of $c_n(t)$ from equation (18), our focus will be on how to calculate the function $c : \mathbb{R}_{\geq 0} \to \mathbb{R}^N$ from the input function $f : \mathbb{R}_{\geq 0} \to \mathbb{R}$.

In our framework, we will compute these coefficients over time by viewing them as a dynamical system. Differentiating (18),

$$
\begin{aligned}
\frac{d}{dt} c_n(t) = {} & \zeta(t)^{-\frac{1}{2}} \lambda_n \int f(x) \left( \frac{\partial}{\partial t} p_n(t,x) \right) \frac{\omega}{\chi}(t,x) \mathrm{d}x \\
& + \int f(x) \left( \zeta^{-\frac{1}{2}} \lambda_n p_n(t,x) \right) \left( \frac{\partial}{\partial t} \frac{\omega}{\chi}(t,x) \right) \mathrm{d}x.
\end{aligned}
\tag{20}
$$

Here we have made use of the assumption that $\zeta$ is constant for all $t$.

Let $c(t) \in \mathbb{R}^{N-1}$ denote the vector of all coefficients $(c_n(t))_{0 \leq n < N}$.

The key idea is that if $\frac{\partial}{\partial t} P_n$ and $\frac{\partial}{\partial t} \frac{\omega}{\chi}$ have closed forms that can be related back to the polynomials $P_k$, then an ordinary differential equation can be written for $c(t)$. This allows these coefficients $c(t)$ and hence the optimal polynomial approximation to be computed online. Since $\frac{d}{dt} P_n^{(t)}$ is a polynomial (in $x$) of degree $n-1$, it can be written as linear combinations of $P_0, ..., P_{n-1}$, so the first term in Eq. (20) is a linear combination of $c_0, ..., c_{n-1}$. For many weight functions $\omega$, we can find scaling function $\chi$ such that $\frac{\partial}{\partial t} \frac{\omega}{\chi}$ can also be written in terms of $\frac{\omega}{\chi}$ itself, and thus in those cases the second term of Eq. (20) is also a linear combination of $c_0, ..., c_{N-1}$ and the input $f$. Thus this often yields a closed-form linear ODE for $c(t)$.

**Normalized dynamics**   Our purpose of defining the free parameters $\lambda_n$ was threefold.

1. First, note that the orthonormal basis is not unique, up to a $\pm 1$ factor per element.
2. Second, choosing $\lambda_n$ can help simplify the derivations.
3. Third, although choosing $\lambda_n = \pm 1$ will be our default, since projecting onto an orthonormal basis is most sensible, the LMU [71] used a different scaling. Appendix D.1 will recover the LMU by choosing different $\lambda_n$ for the LegT measure.

Suppose that equation (20) reduced to dynamics of the form

$$
\frac{d}{dt} c(t) = -A(t) c(t) + B(t) f(t).
$$

Then, letting $\Lambda = \mathrm{diag}_{n \in [N]} \{ \lambda_n \}$,

$$
\frac{d}{dt} \Lambda^{-1} c(t) = -\Lambda^{-1} A(t) \Lambda \Lambda^{-1} c(t) + \Lambda^{-1} B(t) f(t).
$$

Therefore, if we reparameterize the coefficients $(\Lambda^{-1} c(t) \to c(t))$ then the *normalized* coefficients projected onto the orthonormal basis satisfy dynamics and associated reconstruction

$$
\frac{d}{dt} c(t) = -(\Lambda^{-1} A(t) \Lambda) c(t) + (\Lambda^{-1} B(t)) f(t)
\tag{21}
$$

$$
f_{\leq t} \approx g^{(t)} = \sum_{n=0}^{N-1} \zeta^{\frac{1}{2}} c_n(t) p_n^{(t)} \chi^{(t)}
\tag{22}
$$

These are the hippo and $\mathrm{proj}_t$ operators.

### C.4 Discretization

As defined here, hippo is a map on continuous functions. However, as hippo defines a closed-form ODE of the coefficient dynamics, standard ODE discretization methods (Appendix B.3) can be applied

Figure 5: **Illustration of HiPPO measures.** At time $t_0$, the history of a function $f(x)_{x \leq t_0}$ is summarized by polynomial approximation with respect to the measure $\mu^{(t_0)}$ (blue), and similarly for time $t_1$ (purple). (Left) The Translated Legendre measure (**LegT**) assigns weight in the window $[t-\theta, t]$. For small $t$, $\mu^{(t)}$ is supported on a region $x < 0$ where $f$ is not defined. When $t$ is large, the measure is not supported near 0, causing the projection of $f$ to forget the beginning of the function. (Middle) The Translated Laguerre (**LagT**) measure decays the past exponentially. It does not forget, but also assigns weight on $x < 0$. (Right) The Scaled Legendre measure (**LegS**) weights the entire history $[0, t]$ uniformly.

to turn this into discrete memory updates. Thus we overload these operators, i.e. $\mathrm{hippo}$ either defines an ODE of the form

$$\frac{d}{dt}c(t) = A(t)c(t) + B(t)f(t)$$

or a recurrence

$$c_t = A_t c_{t-1} + B_t f_t,$$

whichever is clear from context.

Appendix F.5 validates the framework by applying (20) and (19) to approximate a synthetic function.

# D    Derivations of HiPPO Projection Operators

We derive the memory updates associated with the translated Legendre (LegT) and translated Laguerre (LagT) measures as presented in Section 2.3, along with the scaling Legendre (LegS) measure (Section 3). To show the generality of the framework, we also derive memory updates with Fourier basis (recovering the Fourier Recurrent Unit [79]) and with Chebyshev basis.

The majority of the work has already been accomplished by setting up the projection framework, and the proof simply requires following the technical outline laid out in Appendix C. In particular, the definition of the coefficients (18) and reconstruction (19) does not change, and we only consider how to calculate the coefficients dynamics (20).

For each case, we follow the general steps:

**Measure and Basis**  define the measure $\mu^{(t)}$ or weight $\omega(t,x)$ and basis functions $p_n(t,x)$,

**Derivatives**  compute the derivatives of the measure and basis functions,

**Coefficient Dynamics**  plug them into the coefficient dynamics (equation (20)) to derive the ODE that describes how to compute the coefficients $c(t)$,

**Reconstruction**  provide the complete formula to reconstruct an approximation to the function $f_{\leq t}$, which is the optimal projection under this measure and basis.

The derivations in Appendices D.1 and D.2 prove Theorem 1, and the derivations in Appendix D.3 prove Theorem 2. Appendices D.4 and D.5 show additional results for Fourier-based bases.

Figure 5 illustrates the overall framework when we use Legendre and Laguerre polynomials as the basis, contrasting our main families of time-varying measures $\mu^{(t)}$.

## D.1    Derivation for Translated Legendre (HiPPO-LegT)

This measure fixes a window length $\theta$ and slides it across time.

**Measure and Basis** We use a uniform weight function supported on the interval $[t-\theta,t]$ and pick Legendre polynomials $P_n(x)$, translated from $[-1,1]$ to $[t-\theta,t]$, as basis functions:

$$\omega(t,x) = \frac{1}{\theta} \mathbb{I}_{[t-\theta,t]}$$

$$p_n(t,x) = (2n+1)^{1/2} P_n\left(\frac{2(x-t)}{\theta}+1\right)$$

$$g_n(t,x) = \lambda_n p_n(t,x).$$

Here, we have used no tilting so $\chi = 1$ and $\zeta = 1$ (equations (15) and (16)). We leave $\lambda_n$ unspecified for now.

At the endpoints, these basis functions satisfy

$$g_n(t,t) = \lambda_n (2n+1)^{\frac{1}{2}}$$

$$g_n(t,t-\theta) = \lambda_n (-1)^n (2n+1)^{\frac{1}{2}}.$$

**Derivatives** The derivative of the measure is

$$\frac{\partial}{\partial t}\omega(t,x) = \frac{1}{\theta}\delta_t - \frac{1}{\theta}\delta_{t-\theta}.$$

The derivative of Legendre polynomials can be expressed as linear combinations of other Legendre polynomials (cf. Appendix B.1.1).

$$\frac{\partial}{\partial t}g_n(t,x) = \lambda_n(2n+1)^{\frac{1}{2}} \cdot \frac{-2}{\theta} P_n'\left(\frac{2(x-t)}{\theta}+1\right)$$

$$= \lambda_n(2n+1)^{\frac{1}{2}}\frac{-2}{\theta}\left[(2n-1)P_{n-1}\left(\frac{2(x-t)}{\theta}+1\right)+(2n-5)P_{n-3}\left(\frac{2(x-t)}{\theta}+1\right)+...\right]$$

$$= -\lambda_n(2n+1)^{\frac{1}{2}}\frac{2}{\theta}\left[\lambda_{n-1}^{-1}(2n-1)^{\frac{1}{2}}g_{n-1}(t,x)+\lambda_{n-3}^{-1}(2n-3)^{\frac{1}{2}}g_{n-3}(t,x)+...\right].$$

We have used equation (7) here.

**Sliding Approximation** As a special case for the LegT measure, we need to consider an approximation due to the nature of the sliding window measure.

When analyzing $\frac{d}{dt}c(t)$ in the next section, we will need to use the value $f(t-\theta)$. However, at time $t$ this input is no longer available. Instead, we need to rely on our compressed representation of the function: by the reconstruction equation (19), if the approximation is succeeding so far, we should have

$$f_{\leq t}(x) \approx \sum_{k=0}^{N-1} \lambda_k^{-1} c_k(t)(2k+1)^{\frac{1}{2}} P_k\left(\frac{2(x-t)}{\theta}+1\right)$$

$$f(t-\theta) \approx \sum_{k=0}^{N-1} \lambda_k^{-1} c_k(t)(2k+1)^{\frac{1}{2}}(-1)^k.$$

**Coefficient Dynamics** We are ready to derive the coefficient dynamics.

Plugging the derivatives of this measure and basis into equation (20) gives

$$
\begin{aligned}
\frac{d}{dt}c_n(t) &= \int f(x)\left(\frac{\partial}{\partial t}g_n(t,x)\right)\omega(t,x)\mathrm{d}x \\
&\quad + \int f(x)g_n(t,x)\left(\frac{\partial}{\partial t}\omega(t,x)\right)\mathrm{d}x \\
&= -\lambda_n(2n+1)^{\frac12}\frac{2}{\theta}\left[\lambda_{n-1}^{-1}(2n-1)^{\frac12}c_{n-1}(t)+\lambda_{n-3}^{-1}(2n-5)^{\frac12}c_{n-3}(t)+...\right] \\
&\quad + \frac{1}{\theta}f(t)g_n(t,t)-\frac{1}{\theta}f(t-\theta)g_n(t,t-\theta) \\
&\approx -\frac{\lambda_n}{\theta}(2n+1)^{\frac12}\cdot 2\left[(2n-1)^{\frac12}\frac{c_{n-1}(t)}{\lambda_{n-1}}+(2n-5)^{\frac12}\frac{c_{n-3}(t)}{\lambda_{n-3}}+...\right] \\
&\quad + (2n+1)^{\frac12}\frac{\lambda_n}{\theta}f(t)-(2n+1)^{\frac12}\frac{\lambda_n}{\theta}(-1)^n\sum_{k=0}^{N-1}(2k+1)^{\frac12}\frac{c_k(t)}{\lambda_k}(-1)^k \\
&= -\frac{\lambda_n}{\theta}(2n+1)^{\frac12}\cdot 2\left[(2n-1)^{\frac12}\frac{c_{n-1}(t)}{\lambda_{n-1}}+(2n-5)^{\frac12}\frac{c_{n-3}(t)}{\lambda_{n-3}}+...\right] \\
&\quad - (2n+1)^{\frac12}\frac{\lambda_n}{\theta}\sum_{k=0}^{N-1}(-1)^{n-k}(2k+1)^{\frac12}\frac{c_k(t)}{\lambda_k}+(2n+1)^{\frac12}\frac{\lambda_n}{\theta}f(t) \\
&= -\frac{\lambda_n}{\theta}(2n+1)^{\frac12}\sum_{k=0}^{N-1}M_{nk}(2k+1)^{\frac12}\frac{c_k(t)}{\lambda_k}+(2n+1)^{\frac12}\frac{\lambda_n}{\theta}f(t),
\end{aligned}
$$

where

$$
M_{nk}=\begin{cases}1 & \text{if } k\le n \\ (-1)^{n-k} & \text{if } k\ge n\end{cases}.
$$

Now we consider two instantiations for $\lambda_n$. The first one is the more natural $\lambda_n=1$, which turns $g_n$ into an orthonormal basis. We then get

$$
\frac{d}{dt}c(t)=-\frac{1}{\theta}Ac(t)+\frac{1}{\theta}Bf(t)
$$

$$
A_{nk}=(2n+1)^{\frac12}(2k+1)^{\frac12}\begin{cases}1 & \text{if } k\le n \\ (-1)^{n-k} & \text{if } k\ge n\end{cases}
$$

$$
B_n=(2n+1)^{\frac12}.
$$

The second case takes $\lambda_n=(2n+1)^{\frac12}(-1)^n$. This yields

$$
\frac{d}{dt}c(t)=-\frac{1}{\theta}Ac(t)+\frac{1}{\theta}Bf(t)
$$

$$
A_{nk}=(2n+1)\begin{cases}(-1)^{n-k} & \text{if } k\le n \\ 1 & \text{if } k\ge n\end{cases}
$$

$$
B_n=(2n+1)(-1)^n.
$$

This is exactly the LMU update equation.

**Reconstruction**  By equation (19), at every time $t$ we have

$$
f(x)\approx g^{(t)}(x)=\sum_n \lambda_n^{-1}c_n(t)(2n+1)^{\frac12}P_n\left(\frac{2(x-t)}{\theta}+1\right).
$$

### D.2  Derivation for Translated Laguerre (HiPPO-LagT)

We consider measures based on the generalized *Laguerre* polynomials. For a fixed $\alpha\in\mathbb{R}$, these polynomials $L^{(\alpha)}(t-x)$ are orthogonal with respect to the measure $x^\alpha e^{-x}$ on $[0,\infty)$ (cf. Appendix B.1.2). This derivation will involve tilting the measure governed by another parameter $\beta$.

The result in Theorem 1 for HiPPO-LagT is for the case $\alpha = 0, \beta = 1$, corresponding to the basic Laguerre polynomials and no tilting.

**Measure and Basis**   We flip and translate the generalized Laguerre weight function and polynomials from $[0,\infty)$ to $(-\infty, t]$. The normalization is found using equation (9).

$$\omega(t,x) = \begin{cases} (t-x)^\alpha e^{x-t} & \text{if } x \leq t \\ 0 & \text{if } x > t \end{cases}$$

$$= (t-x)^\alpha e^{-(t-x)} \mathbb{I}_{(-\infty,t]}$$

$$p_n(t,x) = \frac{\Gamma(n+1)^{\frac{1}{2}}}{\Gamma(n+\alpha+1)^{\frac{1}{2}}} L_n^{(\alpha)}(t-x)$$

**Tilted Measure**   We choose the following tilting $\chi$

$$\chi(t,x) = (t-x)^\alpha \exp\left(-\frac{1-\beta}{2}(t-x)\right) \mathbb{I}_{(-\infty,t]}$$

for some fixed $\beta \in \mathbb{R}$. The normalization is (constant across all $t$)

$$\zeta = \int \frac{\omega}{\chi^2} = \int (t-x)^{-\alpha} e^{-\beta(t-x)} \mathbb{I}_{(-\infty,t]} dx$$

$$= \Gamma(1-\alpha)\beta^{\alpha-1},$$

so the tilted measure has density

$$\zeta(t)^{-1}\frac{\omega^{(t)}}{(\chi^{(t)})^2} = \Gamma(1-\alpha)^{-1}\beta^{1-\alpha}(t-x)^{-\alpha}\exp(-\beta(t-x))\mathbb{I}_{(-\infty,t]}.$$

We choose

$$\lambda_n = \frac{\Gamma(n+\alpha+1)^{\frac{1}{2}}}{\Gamma(n+1)^{\frac{1}{2}}}$$

to be the norm of the generalized Laguerre polynomial $L_n^{(\alpha)}$, so that $\lambda_n p_n^{(t)} = L_n^{(\alpha)}(t-x)$, and (following equation (17)) the basis for $\nu^{(t)}$ is

$$g_n^{(t)} = \lambda_n \zeta^{\frac{1}{2}} p_n^{(t)} \chi^{(t)} \tag{23}$$

$$= \zeta^{\frac{1}{2}} \chi^{(t)} L_n^{(\alpha)}(t-x)$$

**Derivatives**   We first calculate the density ratio

$$\frac{\omega}{\chi}(t,x) = \exp\left(-\frac{1+\beta}{2}(t-x)\right)\mathbb{I}_{(-\infty,t]}.$$

and its derivative

$$\frac{\partial}{\partial t}\frac{\omega}{\chi}(t,x) = -\left(\frac{1+\beta}{2}\right)\frac{\omega}{\chi}(t,x) + \exp\left(-\left(\frac{1+\beta}{2}\right)(t-x)\right)\delta_t.$$

The derivative of Laguerre polynomials can be expressed as linear combinations of other Laguerre polynomials (cf. Appendix B.1.2).

$$\frac{\partial}{\partial t}\lambda_n p_n(t,x) = \frac{\partial}{\partial t}L_n^{(\alpha)}(t-x)$$

$$= -L_0^{(\alpha)}(t-x) - \cdots - L_{n-1}^{(\alpha)}(t-x)$$

$$= -\lambda_0 p_0(t,x) - \cdots - \lambda_{n-1}p_{n-1}(t,x)$$

**Coefficient Dynamics** Plugging these derivatives into equation (20) (obtained from differentiating the coefficient equation (18)), where we suppress the dependence on $x$ for convenience:

$$\frac{d}{dt}c_n(t) = \zeta^{-\frac{1}{2}}\int f\cdot\left(\frac{\partial}{\partial t}\lambda_n p_n^{(t)}\right)\frac{\omega^{(t)}}{\chi^{(t)}}$$

$$+\int f\cdot\left(\zeta^{-\frac{1}{2}}\lambda_n p_n^{(t)}\right)\left(\frac{\partial}{\partial t}\frac{\omega^{(t)}}{\chi^{(t)}}\right)$$

$$=-\sum_{k=0}^{n-1}\int f\cdot\left(\zeta^{-\frac{1}{2}}\lambda_k p_k^{(t)}\chi^{(t)}\right)\frac{\omega^{(t)}}{(\chi^{(t)})^2}$$

$$-\left(\frac{1+\beta}{2}\right)\int f\cdot\left(\zeta^{-\frac{1}{2}}\lambda_n p_n^{(t)}\right)\frac{\omega^{(t)}}{\chi^{(t)}}+f(t)\cdot\zeta^{-\frac{1}{2}}L_n^{(\alpha)}(0)$$

$$=-\sum_{k=0}^{n-1}c_k(t)-\left(\frac{1+\beta}{2}\right)c_n(t)+\Gamma(1-\alpha)^{-\frac{1}{2}}\beta^{\frac{1-\alpha}{2}}\binom{n+\alpha}{n}f(t).$$

We then get

$$\frac{d}{dt}c(t)=-Ac(t)+Bf(t)$$

$$A=\begin{bmatrix}\frac{1+\beta}{2} & 0 & \dots & 0\\ 1 & \frac{1+\beta}{2} & \dots & 0\\ \vdots & & \ddots & \\ 1 & 1 & \dots & \frac{1+\beta}{2}\end{bmatrix}$$

$$B=\zeta^{-\frac{1}{2}}\cdot\begin{bmatrix}\binom{\alpha}{0}\\ \vdots\\ \binom{N-1+\alpha}{N-1}\end{bmatrix}$$

(24)

**Reconstruction** By equation (19), at every time $t$, for $x\leq t$,

$$f(x)\approx g^{(t)}(x)=\sum_{n=0}^{N-1}\lambda_n^{-1}\zeta^{\frac{1}{2}}c_n(t)p_n^{(t)}\chi^{(t)}$$

$$=\sum_n\frac{n!}{(n+\alpha)!}\zeta^{\frac{1}{2}}c_n(t)\cdot L_n^{(\alpha)}(t-x)\cdot(t-x)^\alpha e^{\left(\frac{\beta-1}{2}\right)(t-x)}.$$

**Normalized Dynamics** Finally, following equations (21) and (22) to convert these to dynamics on the orthonormal basis of the normalized (probability) measure $\nu^{(t)}$ leads to the following hippo operator

$$\frac{d}{dt}c(t)=-Ac(t)+Bf(t)$$

$$A=-\Lambda^{-1}\begin{bmatrix}\frac{1+\beta}{2} & 0 & \dots & 0\\ 1 & \frac{1+\beta}{2} & \dots & 0\\ \vdots & & \ddots & \\ 1 & 1 & \dots & \frac{1+\beta}{2}\end{bmatrix}\Lambda$$

$$B=\Gamma(1-\alpha)^{-\frac{1}{2}}\beta^{\frac{1-\alpha}{2}}\cdot\Lambda^{-1}\begin{bmatrix}\binom{\alpha}{0}\\ \vdots\\ \binom{N-1+\alpha}{N-1}\end{bmatrix}$$

$$\Lambda=\operatorname*{diag}_{n\in[N]}\left\{\frac{\Gamma(n+\alpha+1)^{\frac{1}{2}}}{\Gamma(n+1)^{\frac{1}{2}}}\right\}$$

(25)

and correspondingly a $\operatorname{proj}_t$ operator:

$$f(x)\approx g^{(t)}(x)=\Gamma(1-\alpha)^{\frac{1}{2}}\beta^{-\frac{1-\alpha}{2}}\sum_n c_n(t)\cdot\frac{\Gamma(n+1)^{\frac{1}{2}}}{\Gamma(n+\alpha+1)^{\frac{1}{2}}}\cdot L_n^{(\alpha)}(t-x)\cdot(t-x)^\alpha e^{\left(\frac{\beta-1}{2}\right)(t-x)}. \quad (26)$$

### D.3 Derivation for Scaled Legendre (HiPPO-LegS)

As discussed in Section 3, the scaled Legendre is our only method that uses a measure with varying width.

**Measure and Basis**   We instantiate the framework in the case

$$\omega(t,x) = \frac{1}{t}\mathbb{I}_{[0,t]} \tag{27}$$

$$g_n(t,x) = p_n(t,x) = (2n+1)^{\frac{1}{2}} P_n\left(\frac{2x}{t}-1\right) \tag{28}$$

Here, $P_n$ are the basic Legendre polynomials (Appendix B.1.1). We use no tilting, i.e. $\chi(t,x)=1$, $\zeta(t)=1$, and $\lambda_n=1$ so that the functions $g_n(t,x)$ are an orthonormal basis.

**Derivatives**   We first differentiate the measure and basis:

$$\frac{\partial}{\partial t}\omega(t,\cdot) = -t^{-2}\mathbb{I}_{[0,t]} + t^{-1}\delta_t = t^{-1}(-\omega(t)+\delta_t)$$

$$\frac{\partial}{\partial t}g_n(t,x) = -(2n+1)^{\frac{1}{2}}2xt^{-2}P_n'\left(\frac{2x}{t}-1\right)$$

$$= -(2n+1)^{\frac{1}{2}}t^{-1}\left(\frac{2x}{t}-1+1\right)P_n'\left(\frac{2x}{t}-1\right).$$

Now define $z=\frac{2x}{t}-1$ for shorthand and apply the properties of derivatives of Legendre polynomials (equation (8)).

$$\frac{\partial}{\partial t}g_n(t,x) = -(2n+1)^{\frac{1}{2}}t^{-1}(z+1)P_n'(z)$$

$$= -(2n+1)^{\frac{1}{2}}t^{-1}\left[nP_n(z)+(2n-1)P_{n-1}(z)+(2n-3)P_{n-2}(z)+...\right]$$

$$= -t^{-1}(2n+1)^{\frac{1}{2}}\left[n(2n+1)^{-\frac{1}{2}}g_n(t,x)+(2n-1)^{\frac{1}{2}}g_{n-1}(t,x)+(2n-3)^{\frac{1}{2}}g_{n-2}(t,x)+...\right]$$

**Coefficient Dynamics**   Plugging these into (20), we obtain

$$\frac{d}{dt}c_n(t) = \int f(x)\left(\frac{\partial}{\partial t}g_n(t,x)\right)\omega(t,x)\mathrm{d}x + \int f(x)g_n(t,x)\left(\frac{\partial}{\partial t}\omega(t,x)\right)\mathrm{d}x$$

$$= -t^{-1}(2n+1)^{\frac{1}{2}}\left[n(2n+1)^{-\frac{1}{2}}c_n(t)+(2n-1)^{\frac{1}{2}}c_{n-1}(t)+(2n-3)^{\frac{1}{2}}c_{n-2}(t)+...\right]$$

$$\quad -t^{-1}c_n(t)+t^{-1}f(t)g_n(t,t)$$

$$= -t^{-1}(2n+1)^{\frac{1}{2}}\left[(n+1)(2n+1)^{-\frac{1}{2}}c_n(t)+(2n-1)^{\frac{1}{2}}c_{n-1}(t)+(2n-3)^{\frac{1}{2}}c_{n-2}(t)+...\right]$$

$$\quad +t^{-1}(2n+1)^{\frac{1}{2}}f(t)$$

where we have used $g_n(t,t)=(2n+1)^{\frac{1}{2}}P_n(1)=(2n+1)^{\frac{1}{2}}$. Vectorizing this yields equation (3):

$$\frac{d}{dt}c(t) = -\frac{1}{t}Ac(t)+\frac{1}{t}Bf(t) \tag{29}$$

$$A_{nk} = \begin{cases} (2n+1)^{1/2}(2k+1)^{1/2} & \text{if } n>k \\ n+1 & \text{if } n=k, \\ 0 & \text{if } n<k \end{cases}$$

$$B_n = (2n+1)^{\frac{1}{2}}$$

Alternatively, we can write this as

$$\frac{d}{dt}c(t) = -t^{-1}D\left[MD^{-1}c(t)+\mathbf{1}f(t)\right], \tag{30}$$

where $D:=\mathrm{diag}\left[(2n+1)^{\frac{1}{2}}\right]_{n=0}^{N-1}$, $\mathbf{1}$ is the all ones vector, and the state matrix $M$ is

$$M=\begin{bmatrix}1 & 0 & 0 & 0 & \cdots & 0 \\ 1 & 2 & 0 & 0 & \cdots & 0 \\ 1 & 3 & 3 & 0 & \cdots & 0 \\ 1 & 3 & 5 & 4 & \cdots & 0 \\ \vdots & \vdots & \vdots & \vdots & \ddots & \vdots \\ 1 & 3 & 5 & 7 & \cdots & N\end{bmatrix}, \quad \text{that is,} \quad M_{nk}=\begin{cases}2k+1 & \text{if } k<n \\ k+1 & \text{if } k=n \\ 0 & \text{if } k>n\end{cases}$$

Equation (29) is a linear dynamical system, except dilated by a time-varying factor $t^{-1}$, which arises from the scaled measure.

**Reconstruction**    By equation (19), at every time $t$ we have

$$f(x)\approx g^{(t)}(x)=\sum_n c_n(t)g_n(t,x).$$
$$=\sum_n c_n(t)(2n+1)^{\frac{1}{2}}P_n\left(\frac{2x}{t}-1\right).$$

## D.4    Derivation for Fourier Bases

In the remainder of Appendix D, we consider some additional bases which are analyzable under the HiPPO framework. These use measures and bases related to various forms of the Fourier transform.

### D.4.1    Translated Fourier

Similar to the LMU, the sliding Fourier measure also has a fixed window length $\theta$ parameter and slides it across time.

**Measure**    The Fourier basis $e^{2\pi inx}$ (for $n=0,...,N-1$) can be seen as an orthogonal polynomials basis $z^n$ with respect to the uniform measure on the unit circle $\{z:|z|=1\}$. By a change of variable $z\to e^{2\pi ix}$ (and thus changing the domain from the unit circle to $[0,1]$), we obtain the usual Fourier basis $e^{2\pi inx}$. The complex inner product $\langle f,g\rangle$ is defined as $\int_0^1 f(x)\overline{g(x)}\mathrm{d}x$. Note that the basis $e^{2\pi inx}$ is orthonormal.

For each $t$, we will use a sliding measure uniform on $[t-\theta,t]$ and rescale the basis as $e^{2\pi in\frac{t-x}{\theta}}$ (so they are still orthonormal, i.e., have norm 1):

$$\omega(t,x)=\frac{1}{\theta}\mathbb{I}_{[t-\theta,t]}$$
$$p_n(t,x)=e^{2\pi in\frac{t-x}{\theta}}.$$

We sue no tilting (i.e., $\chi(t,x)=1$).

**Derivatives**

$$\frac{\partial}{\partial t}\omega(t,x)=\frac{1}{\theta}\delta_t-\frac{1}{\theta}\delta_{t-\theta}$$
$$\frac{\partial}{\partial t}p_n(t,x)=\frac{2\pi in}{\theta}e^{2\pi in\frac{t-x}{\theta}}=\frac{2\pi in}{\theta}p_n(t,x).$$

**Coefficient Updates**    Plugging into equation (20) yields

$$\frac{d}{dt}c_n(t)=\frac{2\pi in}{\theta}c_n(t)+\frac{1}{\theta}f(t)p_n(t,t)-\frac{1}{\theta}f(t-\theta)p_n(t,t-\theta)$$
$$=\frac{2\pi in}{\theta}c_n(t)+\frac{1}{\theta}f(t)-\frac{1}{\theta}f(t-\theta).$$

Note that $p_n(t,t) = p_n(t,t-\theta) = 1$. Additionally, we no longer have access to $f(t-\theta)$ at time $t$, but this is implicitly represented in our compressed representation of the function: $f = \sum_{k=0}^{N-1} c_k(t) p_k(t)$. Thus we approximate $f(t-\theta)$ by $\sum_{k=0}^{N-1} c_k(t) p_k(t,t-\theta) = \sum_{k=0}^{N-1} c_k(t)$. Finally, this yields

$$\frac{d}{dt} c_n(t) = \frac{2\pi i n}{\theta} c_n(t) + \frac{1}{\theta} f(t) - \frac{1}{\theta} \sum_{k=0}^{N-1} c_k(t).$$

Hence $\frac{d}{dt} c(t) = Ac(t) + Bf(t)$ where

$$A_{nk} = \begin{cases} -1/\theta & \text{if } k \neq n, \\ (2\pi i n - 1)/\theta & \text{if } k = n \end{cases}, \qquad B_n = \frac{1}{\theta}.$$

**Reconstruction**  At every time step $t$, we have

$$f(x) \approx \sum_n c_n(t) p_n(t,x) = \sum_n c_n(t) e^{2\pi i \frac{t-x}{\theta}}.$$

### D.4.2  Fourier Recurrent Unit

Using the HiPPO framework, we can also derive the Fourier Recurrent Unit (FRU) [79].

**Measure**  For each $t$, we will use a sliding measure uniform on $[t-\theta,t]$ and the basis $e^{2\pi i n \frac{x}{\theta}}$:

$$\omega(t,x) = \frac{1}{\theta} \mathbb{I}_{[t-\theta,t]}$$
$$p_n(t,x) = e^{2\pi i n \frac{x}{\theta}}.$$

In general the basis is not orthogonal with respect to the measure $\omega(t,x)$, but orthogonality holds at the end where $t = \theta$.

**Derivatives**

$$\frac{\partial}{\partial t} \omega(t,x) = \frac{1}{\theta} \delta_t - \frac{1}{\theta} \delta_{t-\theta}$$
$$\frac{\partial}{\partial t} p_n(t,x) = 0.$$

**Coefficient Updates**  Plugging into equation 20 yields

$$\frac{d}{dt} c_n(t) = \frac{1}{\theta} f(t) p_n(t,t) - \frac{1}{\theta} f(t-\theta) p_n(t,t-\theta)$$
$$= \frac{1}{\theta} e^{2\pi i n \frac{t}{\theta}} f(t) - \frac{1}{\theta} e^{2\pi i n \frac{t}{\theta}} f(t-\theta).$$

We no longer have access to $f(t-\theta)$ at time $t$, but we can approximate by ignoring this term (which can be justified by assuming that the function $f$ is only defined on $[0,\theta]$ and thus $f(x)$ can be set to zero for $x < 0$). Finally, this yields

$$\frac{d}{dt} c_n(t) = \frac{e^{2\pi i n \frac{t}{\theta}}}{\theta} f(t).$$

Applying forward Euler discretization (with step size = 1), we obtain:

$$c_n(k+1) = c_n(k) + \frac{e^{2\pi i n \frac{t}{\theta}}}{\theta} f(t).$$

Taking the real parts yields the Fourier Recurrent Unit updates [79].

Note that the recurrence is independent in each $n$, so we don't have the pick $n = 0,1,...,N-1$. We can thus pick random frequencies $n$ as done in Zhang et al. [79].

## D.5 Derivation for Translated Chebyshev

The final family of orthogonal polynomials we analyze under the HiPPO framework are the Chebyshev polynomials. The Chebyshev polynomials can be seen as the purely real analog of the Fourier basis; for example, a Chebyshev series is related to a Fourier cosine series through a change of basis [8].

**Measure and Basis**  The basic Chebyshev measure is $\omega^{\mathrm{cheb}} = (1-x^2)^{-1/2}$ on $(-1,1)$. Following Appendix B.1.3, we choose the following measure and orthonormal basis polynomials in terms of the Chebyshev polynomials of the first kind $T_n$.

$$\omega(t,x) = \frac{2}{\theta\pi}\omega^{\mathrm{cheb}}\left(\frac{2(x-t)}{\theta}+1\right)\mathbb{I}_{(t-\theta,t)}$$

$$= \frac{1}{\theta\pi}\left(\frac{x-t}{\theta}+1\right)^{-1/2}\left(-\frac{x-t}{\theta}\right)^{-1/2}\mathbb{I}_{(t-\theta,t)}$$

$$p_n(t,x) = \sqrt{2}T_n\left(\frac{2(x-t)}{\theta}+1\right) \qquad \text{for } n \geq 1,$$

$$p_0(t,x) = T_0\left(\frac{2(x-t)}{\theta}+1\right).$$

Note that at the endpoints, these evaluate to

$$p_n(t,t) = \begin{cases} \sqrt{2}T_n(1) = \sqrt{2} & n \geq 1 \\ T_n(1) = 1 & n = 0 \end{cases}$$

$$p_n(t,t-\theta) = \begin{cases} \sqrt{2}T_n(-1) = \sqrt{2}(-1)^n & n \geq 1 \\ T_n(-1) = 1 & n = 0 \end{cases}$$

**Tilted Measure**  Now we choose

$$\chi^{(t)} = 8^{-1/2}\theta\pi\omega^{(t)},$$

So

$$\frac{\omega}{\chi^2} = \frac{1}{\frac{\theta^2\pi^2}{8}\omega} = \frac{8}{\theta\pi}\left(\frac{x-t}{\theta}+1\right)^{1/2}\left(-\frac{x-t}{\theta}\right)^{1/2}\mathbb{I}_{(t-\theta,t)}$$

which integrates to 1.

We also choose $\lambda_n = 1$ for the canonical orthonormal basis, so

$$g^{(t)} = p_n^{(t)}\chi^{(t)}$$

**Derivatives**  The derivative of the density is

$$\frac{\partial}{\partial t}\frac{\omega}{\chi} = \frac{\partial}{\partial t}\frac{8^{1/2}}{\theta\pi}\mathbb{I}_{(t-\theta,t)} = \frac{8^{1/2}}{\theta\pi}(\delta_t - \delta_{t-\theta}).$$

We consider differentiating the polynomials separately for $n=0$, $n$ even, and $n$ odd, using equation (11). Defined $z = \frac{2(x-t)}{\theta}+1$ for convenience. First, for $n$ even,

$$\frac{\partial}{\partial t}p_n(t,x) = -\frac{2^{\frac{3}{2}}}{\theta}T'_n\left(\frac{2(x-t)}{\theta}+1\right)$$

$$= -\frac{2^{\frac{3}{2}}}{\theta}T'_n(z)$$

$$= -\frac{2^{\frac{3}{2}}}{\theta}\cdot 2n(T_{n-1}(z)+T_{n-3}(z)+\cdots+T_1(z))$$

$$= -\frac{4n}{\theta}(p_{n-1}(t,x)+p_{n-3}(t,x)+\cdots+p_1(t,x))$$

For $n$ odd,

$$\frac{\partial}{\partial t}p_n(t,x) = -\frac{2^{\frac{3}{2}}}{\theta}T_n'\left(\frac{2(x-t)}{\theta}+1\right)$$

$$= -\frac{2^{\frac{3}{2}}}{\theta}T_n'(z)$$

$$= -\frac{2^{\frac{3}{2}}}{\theta}\cdot 2n\left(T_{n-1}(z)+T_{n-3}(z)+\cdots+T_1(z)+\frac{1}{2}T_0(z)\right)$$

$$= -\frac{4n}{\theta}\left(p_{n-1}(t,x)+p_{n-3}(t,x)+\cdots+2^{-\frac{1}{2}}p_0(t,x)\right)$$

And

$$\frac{\partial}{\partial t}p_0(t,x)=0.$$

**Coefficient Dynamics**

$$c_n(t)=\int f(x)p_n(t,x)\frac{2^{3/2}}{\theta\pi}\mathbb{I}_{(t-\theta,t)}\mathrm{d}x$$

$$\frac{d}{dt}c_n(t)=\int f(x)\frac{\partial}{\partial t}p_n(t,x)\frac{2^{3/2}}{\theta\pi}\mathbb{I}_{(t-\theta,t)}\mathrm{d}x+\frac{2^{3/2}}{\theta\pi}f(t)p_n(t,t)-\frac{2^{3/2}}{\theta\pi}f(t-\theta)p_n(t,t-\theta)$$

$$= -\frac{4n}{\theta}(c_{n-1}+c_{n-3}+...)+\frac{2^{3/2}}{\theta\pi}f(t)\begin{cases}\sqrt{2} & n\geq 1\\ 1 & n=0\end{cases},$$

where we take $f(t-\theta)=0$ as we no longer have access to it (this holds when $t<\theta$ as well).

In the usual way, we can write this as linear dynamics

$$\frac{d}{dt}c(t)=-\frac{1}{\theta}Ac(t)+\frac{1}{\theta}Bf(t)$$

$$A=4\begin{bmatrix}0 & & & & \cdots\\ 2^{-\frac{1}{2}} & 0 & & & \\ 0 & 2 & 0 & & \cdots\\ 2^{-\frac{1}{2}}\cdot 3 & 0 & 3 & 0 & \\ & \ddots & & \ddots & \end{bmatrix}$$

$$B=\frac{2^{3/2}}{\pi}\begin{bmatrix}1\\ \sqrt{2}\\ \sqrt{2}\\ \sqrt{2}\\ \vdots\end{bmatrix}$$

**Reconstruction** In the interval $(t-\theta,t)$,

$$f(x)\approx\sum_{n=0}^{N-1}c_n(t)p_n(t,x)\chi(t,x).$$

# E  HiPPO-LegS Theoretical Properties

## E.1  Timescale equivariance

*Proof of Proposition 3.* Let $\tilde{f}(t) = f(\alpha t)$. Let $c = \text{proj}\,f$ and $\tilde{c} = \text{proj}\,\tilde{f}$. By the HiPPO equation (18) update and the basis instantiation for LegS (equation (28)),

$$
\begin{aligned}
\tilde{c}_n(t) &= \langle \tilde{f}, g_n^{(t)} \rangle_{\mu^{(t)}} \\
&= \int \tilde{f}(t)(2n+1)^{\frac{1}{2}} P_n\left(2\frac{x}{t}-1\right)\frac{1}{t}\mathbb{I}_{[0,1]}\left(\frac{x}{t}\right)\mathrm{d}x \\
&= \int f(\alpha t)(2n+1)^{\frac{1}{2}} P_n\left(2\frac{x}{t}-1\right)\frac{1}{t}\mathbb{I}_{[0,1]}\left(\frac{x}{t}\right)\mathrm{d}x \\
&= \int f(\alpha t)(2n+1)^{\frac{1}{2}} P_n\left(2\frac{x}{\alpha t}-1\right)\frac{1}{\alpha t}\mathbb{I}_{[0,1]}\left(\frac{x}{\alpha t}\right)\mathrm{d}x \\
&= c_n(\alpha t).
\end{aligned}
$$

The second-to-last equality uses the change of variables $x \mapsto \frac{x}{\alpha}$. $\hspace{2cm}\square$

## E.2  Speed

In this section we work out the fast update rules according to the forward Euler, backward Euler, bilinear, or generalized bilinear transform discretizations (cf. Appendix B.3). Recall that we must be able to perform matrix-vector multiplication by $I + \delta A$ and $(I - \delta A)^{-1}$ where $\delta$ is some multiple of the step size $\Delta t$ (equation (13)).

It is easily seen that the LegS update rule involves a matrix $A$ of the following form (Theorem 2): $A = D_1(L + D_0)D_2$, where $L$ is the all 1 lower triangular matrix and $D_0, D_1, D_2$ are diagonal. Clearly, $I + \delta A$ is efficient (only requiring $O(N)$ operations), as it only involves matrix-vector multiplication by diagonals $D_0, D_1, D_2$, or multiplication by $L$ which is the cumsum operation.

Now we consider multiplication by the inverse $(I + \delta A)^{-1}$ (the minus sign can be absorbed into $\delta$). Write

$$
\begin{aligned}
(I + \delta D_1(L+D_0)D_2)^{-1} &= \left(D_1(D_1^{-1}D_2^{-1} + \delta(L+D_0))D_2\right)^{-1} \\
&= \delta^{-1}D_2^{-1}\left(\delta^{-1}D_1^{-1}D_2^{-1} + D_0 + L\right)^{-1}D_1^{-1}
\end{aligned}
$$

Since diagonal multiplication is efficient, the crucial operation is inversion multiplication by a matrix of the form $L + D$.

Consider solving the equation $(L+D)x = y$. This implies $x_0 + \cdots + x_{k-1} = y_k - (1+d_k)x_k$. The solution is

$$
x_0 = \frac{y_0}{1+d_0}
$$

$$
x_k = \frac{y_k - x_0 - \cdots - x_{k-1}}{1+d_k}
$$

Define $s_k = x_0 + \cdots + x_k$. Then

$$
s_k = s_{k-1} + x_k = s_{k-1} + \frac{y_k - s_{k-1}}{1+d_k} = \frac{y_k + d_k s_{k-1}}{1+d_k} = \frac{d_k}{1+d_k}s_{k-1} + \frac{y_k}{1+d_k}.
$$

Finally, consider how to calculate a recurrence of the following form efficiently.

$$
x_0 = \beta_0, x_k = \alpha_k x_{k-1} + \beta_k.
$$

This update rule can also be written

$$
\frac{x_k}{\alpha_k \ldots \alpha_1} = \frac{x_{k-1}}{\alpha_{k-1}\ldots\alpha_1} + \frac{\beta_k}{\alpha_k\ldots\alpha_1}.
$$

Evidently $x$ can be computed in a vectorized way as

$$
x = \mathsf{cumsum}(\beta/\mathsf{cumprod}(\alpha))\cdot\mathsf{cumprod}(\alpha).
$$

This is an $O(N)$ computation.

### E.3 Gradient Norms

We analyze the discrete time case under the Euler discretization (Appendix B.3), where the HiPPO-LegS recurrent update is equation (4), restated here for convenience:

$$c_{k+1} = \left(1 - \frac{A}{k}\right)c_k + \frac{1}{k}Bf_k.$$

These gradient asymptotics hold under other discretizations.

We will show that

**Proposition 7.** *For any times $k < \ell$, the gradient norm of the HiPPO-LegS operator for the output at time $\ell+1$ with respect to input at time $k$ is $\left\|\frac{\partial c_{\ell+1}}{\partial f_k}\right\| = \Theta(1/\ell)$.*

*Proof.* We take $N$ to be a constant.

Without loss of generality assume $k > 2$, as the gradient change for a single initial step is bounded. By unrolling the recurrence (4), the dependence of $c_{\ell+1}$ on $c_k$ and $f_k,...,f_\ell$ can be made explicit:

$$
\begin{aligned}
c_{\ell+1} = &\left(I - \frac{A}{\ell}\right)...\left(I - \frac{A}{k}\right)c_k \\
&+ \left(I - \frac{A}{\ell}\right)...\left(I - \frac{A}{k+1}\right)\frac{B}{k}f_k \\
&+ \left(I - \frac{A}{\ell}\right)...\left(I - \frac{A}{k+2}\right)\frac{B}{k+1}f_{k+1} \\
&\vdots \\
&+ \left(I - \frac{A}{\ell}\right)\frac{B}{\ell-1}f_{\ell-1} \\
&+ \frac{B}{\ell}f_\ell.
\end{aligned}
$$

Therefore

$$\frac{\partial c_{\ell+1}}{\partial f_k} = \left(I - \frac{A}{\ell}\right)...\left(I - \frac{A}{k+1}\right)\frac{B}{k}.$$

Notice that $A$ has distinct eigenvalues $1,2,...,N$, since those are the elements of its diagonal and $A$ is triangular (Theorem 2). Thus the matrices $I - \frac{A}{\ell},...,I - \frac{A}{k+1}$ are diagonalizable with a common change of basis. The gradient then has the form $PDP^{-1}B$ for some invertible matrix $P$ and some diagonal matrix $D$. Its norm is therefore bounded from below (up to constant) by the smallest singular value of $P$ and $\|P^{-1}B\|$, both of which are nonzero constants, and the largest diagonal entry of $D$. It thus suffices to bound this largest diagonal entry of $D$, which is the largest eigenvalue of this product,

$$\rho = \left(1 - \frac{1}{\ell}\right)...\left(1 - \frac{1}{k+1}\right)\frac{1}{k}.$$

The problem reduces to showing that $\rho = \Theta(1/l)$.

We will use the following facts about the function $\log\left(1 - \frac{1}{x}\right)$. First, it is an increasing function, so

$$\log\left(1 - \frac{1}{x}\right) \geq \int_{x-1}^{x} \log\left(1 - \frac{1}{\lambda}\right)d\lambda.$$

Second, its antiderivative is

$$\int \log\left(1 - \frac{1}{x}\right) = \int \log(x-1) - \log(x) = (x-1)\log(x-1) - x\log(x) = x\log\left(1 - \frac{1}{x}\right) - \log(x-1).$$

Therefore, we have

$$\log\left(1-\frac{1}{\ell}\right)\dots\left(1-\frac{1}{k+1}\right) = \sum_{i=k+1}^{\ell}\log\left(1-\frac{1}{i}\right)$$

$$\geq \sum_{i=k+1}^{\ell}\int_{i-1}^{i}\log\left(1-\frac{1}{x}\right)\mathrm{d}x$$

$$= \int_{k}^{\ell}\log\left(1-\frac{1}{x}\right)\mathrm{d}x$$

$$= \left[(x-1)\log(x-1)-x\log(x)\right]\big|_{k}^{\ell}$$

$$= \ell\log\left(1-\frac{1}{\ell}\right)-\log(\ell-1)$$

$$- \left(k\log\left(1-\frac{1}{k}\right)-\log(k-1)\right).$$

Finally, note that $x\log\left(1-\frac{1}{x}\right)$ is an increasing function, and bounded from above since it is negative, so it is $\Theta(1)$ (this can also be seen from its Taylor expansion). Thus we have

$$\log\rho \geq \Theta(1)-\log(\ell-1)+\log(k-1)-\log(k),$$

Furthermore, all inequalities are asymptotically tight, so that $\rho=\Theta(1/\ell)$ as desired.

$\square$

### E.4 Function Approximation Error

*Proof of Proposition 6.* Fix a time $t$. HiPPO-LegS uses the measure $\omega(t,x) = \frac{1}{t}\mathbb{I}_{[0,t]}$ and the polynomial basis $p_n(t,x)=(2n+1)^{\frac{1}{2}}P_n\left(\frac{2x}{t}-1\right)$. Let $c_n(t)=\langle f_{\leq t},p_n^{(t)}\rangle_{\mu^{(t)}}$ for $n=0,1,\dots$. Then the projection $g^{(t)}$ is obtained by linear combinations of the basis functions, with $c_n(t)$ as coefficients:

$$g^{(t)} = \sum_{n=0}^{N-1}c_n(t)p_n^{(t)}.$$

Since $p_n^{(t)}$ forms an orthonormal basis of the Hilbert space defined by the inner product $\langle\cdot,\cdot\rangle_{\mu^{(t)}}$ [14], by Parseval's identity,

$$\left\|f_{\leq t}-g^{(t)}\right\|_{\mu^{(t)}}^2 = \sum_{n=N}^{\infty}c_n^2(t).$$

To bound the error $\left\|f_{\leq t}-g^{(t)}\right\|_{\mu^{(t)}}$, it suffices to bound the sum of the squares of the high-order coefficients $c_n(t)$ for $n=N,N+1,\dots$. We will bound each coefficient by integration by parts.

We first simplify the expression for $c_n(t)$. For any $n\geq 1$, we have

$$c_n(t) = \langle f_{\leq t},p_n^{(t)}\rangle_{\mu^{(t)}}$$

$$= \frac{1}{t}(2n+1)^{\frac{1}{2}}\int_0^t f(x)P_n\left(\frac{2x}{t}-1\right)\mathrm{d}x$$

$$= \frac{(2n+1)^{\frac{1}{2}}}{2}\int_{-1}^1 f\left(\frac{1+x}{2}t\right)P_n(x)\mathrm{d}x \qquad \text{(change of variable } x\to\frac{1+x}{2}t\text{)}.$$

As $P_n(x)=\frac{1}{2n+1}\frac{d}{dx}(P_{n+1}(x)-P_{n-1}(x))$ (cf. Appendix B.1.1), integration by parts yields:

$$c_n(t) = \frac{(2n+1)^{\frac{1}{2}}}{2}\left[f\left(\frac{1+x}{2}t\right)\frac{1}{2n+1}(P_{n+1}(x)-P_{n-1}(x))\right]\Big|_{-1}^1$$

$$- \frac{(2n+1)^{\frac{1}{2}}}{2}\int_{-1}^1\frac{t}{2}f'\left(\frac{1+x}{2}t\right)\frac{1}{2n+1}(P_{n+1}(x)-P_{n-1}(x))\mathrm{d}x.$$

Notice that the boundary term is zero, since $P_{n+1}(1) = P_{n-1}(1) = 1$ and $P_{n+1}(-1) = P_{n-1}(-1) = \pm 1$ (either both 1 or both $-1$ depending on whether $n$ is odd or even). Hence:

$$c_n(t) = -\frac{1}{4} \cdot \frac{1}{(2n+1)^{\frac{1}{2}}} \cdot t \int_{-1}^{1} f'\left(\frac{1+x}{2}t\right)(P_{n+1}(x) - P_{n-1}(x))\mathrm{d}x.$$

Now suppose that $f$ is $L$-Lipschitz, which implies that $|f'| \leq L$. Then

$$c_n^2(t) \leq t^2 L^2 \frac{1}{16} \cdot \frac{1}{2n+1} \left[\int_{-1}^{1} |P_{n+1}(x) - P_{n-1}(x)|\mathrm{d}x\right]^2$$

$$\leq t^2 L^2 \frac{1}{16} \cdot \frac{1}{2n+1} \cdot 2 \int_{-1}^{1} (P_{n+1}(x) - P_{n-1}(x))^2 \mathrm{d}x \qquad \text{(Cauchy–Schwarz)}$$

$$= t^2 L^2 \frac{1}{8} \frac{1}{2n+1} \left[\int_{-1}^{1} P_{n+1}^2(x)\mathrm{d}x + \int_{-1}^{1} P_{n-1}^2(x)\mathrm{d}x\right] \qquad (P_{n+1} \text{ and } P_{n-1} \text{ are orthogonal})$$

$$= t^2 L^2 \frac{1}{8} \frac{1}{2n+1} \left[\frac{2}{2n+3} + \frac{2}{2n-1}\right]$$

$$= O(1)t^2 L^2 \frac{1}{n^2}.$$

Summing for all $n \geq N$ yields:

$$\left\|f_{\leq t} - g^{(t)}\right\|_{\mu^{(t)}}^2 = \sum_{n=N}^{\infty} c_n^2(t) = O(1)t^2 L^2 \sum_{n=N}^{\infty} \frac{1}{n^2} = O(1)t^2 L^2 \frac{1}{N}.$$

We then obtain that $\left\|f_{\leq t} - g^{(t)}\right\|_{\mu^{(t)}} = O(tL/\sqrt{N})$ as claimed.

Now supposed that $f$ has $k$ derivatives and the $k$-th derivative is bounded. The argument is similar to the one above where we integrate by parts $k$ times. We sketch this argument here.

Take $k$ to be a constant, and let $n \geq k$. Applying integration by parts $k$ times, noting that all the boundary terms are zero, gives:

$$c_n(t) = O(1)(2n+1)^{\frac{1}{2}} t^k \int_{-1}^{1} f^{(k)}\left(\frac{1+x}{2}t\right) q_k(x)\mathrm{d}x,$$

where $q_k(x)$ is a polynomial such that $\frac{\mathrm{d}^k}{\mathrm{d}x^k} q_k(x) = P_n(x)$. Then since $f^{(k)}$ is bounded, $|c_n(t)| = O(1)(2n+1)^{\frac{1}{2}} \int_{-1}^{1} |q_k(x)|\mathrm{d}x$, and so

$$c_n^2(t) = O(1)t^{2k}(2n+1)\left[\int_{-1}^{1} |q_k(x)|\mathrm{d}x\right]^2 = O(1)t^{2k}(2n+1)\int_{-1}^{1} q_k^2(x)\mathrm{d}x \quad \text{(Cauchy–Schwarz)}.$$

It remains to bound $\int_{-1}^{1} q_k^2(x)\,\mathrm{d}x$. Using the fact that $\frac{\mathrm{d}}{\mathrm{d}x}P_n(x) = \frac{1}{2n+1}(P_{n+1}(x) - P_{n-1}(x))$ repeatedly, we have:

$$q_1 = \frac{1}{2n+1}(P_{n+1} - P_{n-1}) = \frac{1}{n+O(1)} \cdot \frac{1}{2}(P_{n+1} - P_{n-1})$$

$$q_2 = \frac{1}{(n+O(1))^2}\frac{1}{2^2}(P_{n+2} - P_n - P_n + P_{n-2}) = \frac{1}{(n+O(1))^2}\frac{1}{2^2}(P_{n+2} - 2P_n + P_{n-2})$$

$$q_3 = \frac{1}{(n+O(1))^3}\frac{1}{2^3}(P_{n+3} - P_{n+1} - 2P_{n+1} + 2P_{n-1} + P_{n-1} - P_{n-3}) = \frac{1}{(n+O(1))^3}\frac{1}{2^3}(P_{n+3} - 3P_{n+1} + 3P_{n-1} - P_{n-3})$$

...

In general, when we expand out $\int_{-1}^{1} q_k^2(x)\mathrm{d}x$, since the $P_m$'s are orthogonal, we get $k+1$ terms of the form $\frac{1}{(n+O(1))^{2k}}\frac{1}{2^{2k}}\binom{k}{l}^2 \int_{-1}^{1} P_m^2(x)\mathrm{d}x$ for $k$ different values of $m$ in the range $[n-k, n+k]$, and $l$ goes from 0 to $k$. For each $m$, $\int_{-1}^{1} P_m^2(x)\mathrm{d}x = \frac{1}{n+O(1)}$, and $\sum_{l=0}^{k}\binom{k}{l}^2 = \binom{2k}{k}$. Summing up all $k+1$ terms yields

$$\int_{-1}^{1} q_k^2(x)\mathrm{d}x = \frac{1}{(n+O(1))^{2k+1}}\frac{1}{2^k}\binom{2k}{k}.$$

By Stirling's approximation, $\binom{2k}{k} = O(1)4^k$, so $\int_{-1}^{1} q_k^2(x)\,dx = \frac{O(1)2^k}{(n+O(1))^{2k+1}}$. Noting that $k$ is a constant, plugging this into the bound for $c_n^2(t)$:

$$c_n^2(t) = O(1)t^{2k}(2n+1)\frac{O(1)2^k}{(n+O(1))^{2k+1}} = O(1)t^{2k}\frac{1}{n^{2k}}.$$

Summing for all $n \geq N$ yields:

$$\left\|f_{\leq t} - g^{(t)}\right\|_{\mu^{(t)}}^2 = \sum_{n=N}^{\infty} c_n^2(t) = O(1)t^{2k}\sum_{n=N}^{\infty}\frac{1}{n^{2k}} = O(1)t^{2k}\frac{1}{N^{2k-1}}.$$

We then obtain that $\left\|f_{\leq t} - g^{(t)}\right\|_{\mu^{(t)}} = O(t^k N^{-k+1/2})$ as claimed. □

**Remark.** The approximation error of Legendre polynomials reduces to how fast the Legendre co-efficients decay, subjected to the smoothness assumption of the input function. This result is analogous to the classical result in Fourier analysis, where the $n$-th Fourier coefficients decay as $O(n^{-k})$ if the input function has order-$k$ bounded derivatives [45]. That result is also proved by integration by parts.

## F    Experiment Details and Additional Results

### F.1    Model Architecture Details

Given inputs $x_t$ or features thereof $f(x_t)$ in any model, the HiPPO framework can be used to memorize the history of features $f_t$ through time. As the discretized HiPPO dynamics form a linear recurrent update similar in style to RNNs (e.g., Theorem 2), we focus on these models in our experiments. Thus, given any RNN update function $h_t = \tau(h_{t-1}, x_t)$, we simply replace the previous hidden state with a projected version of its entire history. Equations (31) lists the explicit update equations and Figure 6

$$
\begin{aligned}
h_t \in \mathbb{R}^d &= \tau(h_{t-1}, [c_{t-1}, x_t]) \\
f_t \in \mathbb{R}^1 &= \mathcal{L}_f(h_t) \\
c_t \in \mathbb{R}^N &= \text{hippo}_t(f) \\
&= A_t c_{t-1} + B_t f_t
\end{aligned}
\tag{31}
$$

Figure 6: The simple RNN model we use HiPPO with, and associated update equations. $\mathcal{L}_\square$ is a parametrized linear function, $\tau$ is any RNN update function, and $[\cdot]$ denotes concatenation. hippo is the HiPPO memory operator which orthogonalizes the history of the $f_t$ features up to time $t$. $A_t, B_t$ are fixed matrices depending on the chosen measure . $N$ and $d$ represent the approximation order and hidden state size, respectively.

illustrates the model. In our experiments, we choose a basic gated RNN update

$$\tau(h,x) = (1 - g(h,x)) \circ h + g(h,x) \circ \tanh(\mathcal{L}_\tau(h,x)), \qquad g(h,x) = \sigma(\mathcal{L}_g(h,x)).$$

**Methods and Baselines**    We consider the following instantiations of our framework HiPPO.

**HiPPO-LegT**, **LagT**, and **LegS**, use the translated Legendre, and tilted Laguerre, and scaled Legendre measure families with update dynamics (1), (2), and (3). As mentioned, LegT has an additional hyperparameter $\theta$, which should be set to the timescale of the data if known a priori. We attempt to set it equal to its ideal value (the length of the sequences) in every task, and also consider $\theta$ values that are too large and small to illustrate the effect of this hyperparameter.

Our derivations in Appendices D.1 to D.5 show that there is a large variety of update equations that can arise from the HiPPO framework—for example, the tilted generalized Laguerre polynomials lead to an entire family governed by two free parameters (Appendix D.2)—many of which lead to linear dynamics of the form $\frac{d}{dt}c(t) = -Ac(t) + Bf(t)$ for various $A, B$. Given that many different update dynamics lead to such dynamical systems that give sensible results, we additionally consider the **HiPPO-Rand** baseline that uses random $A$ and $B$ matrices (normalized appropriately) in its dynamics.

We additionally compare against the following standard RNN baselines. The **RNN** is a vanilla RNN. The **MGU** is a minimal gated architecture, equivalent to a **GRU** without the reset gate. The HiPPO architecture we use is simply the MGU with an additional hippo intermediate layer. The **LSTM** is the most well-known and popular RNN architecture, which is a more sophisticated gated RNN. The **expRNN** [48] is the state-of-the-art representative of the *orthogonal RNN* family of models designed for long-term dependencies [3]. The **LMU** is the exact same model as in Voelker et al. [71]; it is equivalent to HiPPO-LegT with a different RNN architecture.

All methods have the same hidden size in our experiments. In particular, for simplicity and to reduce hyperparameters, HiPPO variants tie the memory size $N$ to the hidden state dimension $d$. The hyperparameter $N$ and $d$ is also referred to as the number of hidden units.

**Model and Architecture Comparisons**    The model (31) we use is a simple RNN that bears similarity to the classical LSTM and the original LMU cell. In comparison to the LSTM, HiPPO can be seen as a variant where the memory $m_t$ plays the role of the LSTM's hidden state and $h_t$ plays the role of the LSTM's gated cell state, with equal dimensionalities. HiPPO updates $m_t$ using the fixed $A$ transition matrix instead of a learned matrix, and also lacks "input" and "output" gates, so for a given hidden size, it requires about half the parameters.

The LMU is a version of the HiPPO-LegT cell with an additional hidden-to-hidden transition matrix and memory-to-memory transition vector instead of the gate $g$, leaving it with approximately the same number of trainable parameters.

**Training Details**    Unless stated otherwise, all methods use the Adam optimizer [41] with learning rate frozen to $0.001$, which has been a robust default for RNN based models [31, 71].

All experiments use PyTorch 1.5 and are run on a Nvidia P100 GPU.

### F.2    Permuted MNIST

**Task**    The input to the sequential MNIST (sMNIST) task [47] is an MNIST source image, flattened in row-major order into a single sequence of length 784. The goal of the model is to process the entire image sequentially before outputting a classification label, requiring learning long-term dependencies. A variant of this, the permuted MNIST (pMNIST) task, applies a fixed permutation to every image, breaking locality and further straining a model's capacity for long-term dependencies.

Models are trained using the cross-entropy loss. We use the standard train-test split (60,000 examples for training and 10,000 for testing), and further split the training set with 10% to be used as validation set.

**Baselines and Ablations**    Table 1 is duplicated here in Tables 4 and 5, with more complete baselines and hyperparameter ablations.

Table 4 consists of our implementations of various baselines related to our method, described in Appendix F.1. Each method was ran for 3 seeds, and the maximum average validation accuracy is reported. All methods used the same hidden size of $512$; we found that this gave better performance than 256, and further increasing it did not improve more. All methods were trained for 50 epochs with a batch size of 100.

**State of the Art**    Table 5 directly shows the reported test accuracy of various methods on this data (Middle and Bottom). Table 5 (Top) reports the test accuracy of various instantiations of our methods. We additionally include our reproduction of the LMU, which achieved better results than reported in Voelker et al. [71] (possibly due to a larger hidden size). We note that *all* of our HiPPO methods are competitive; each of them (HiPPO-LegT, HiPPO-LagT, HiPPO-LegS) achieves state-of-the-art among previous recurrent sequence models. Note that differences between our HiPPO-LegT and LMU numbers in Table 5 (Top) stem primarily from the architecture difference (Appendix F.1).

**Timescale Hyperparameters**   Table 4 also shows ablations for the HiPPO-LegT and HiPPO-LagT timescale hyperparameters.   HiPPO-LagT sweeps the discretization step size $\Delta t$ (Section 2.4 and Appendix B.3). For LegT, we set $\Delta t = 1.0$ without loss of generality, as only the ratio of $\theta$ to $\Delta t$ matters. These timescale hyperparameters are important for these methods. Previous works have shown that the equivalent of $\Delta t$ in standard RNNs, i.e. the gates of LSTMs and GRUs (Section 2.4), can also drastically affect their performance [31, 66]. For example, the only difference between the URLSTM and LSTM in Table 5 is a reparametrization of the gates.

Table 4: **Our methods and related baselines**. Permuted MNIST (pMNIST) validation scores. (Top): Our methods. (Bottom): Recurrent baselines.

| Method | Validation accuracy (%) |
|---|---|
| **HiPPO-LegS** | **98.34** |
| HiPPO-LagT $\Delta t = 1.0$ | 98.15 |
| HiPPO-LegT $\theta = 200$ | 98.00 |
| HiPPO-LegT $\theta = 2000$ | 97.90 |
| HiPPO-LagT $\Delta t = 0.1$ | 96.44 |
| HiPPO-LegT $\theta = 20$ | 91.75 |
| HiPPO-LagT $\Delta t = 0.01$ | 90.71 |
| HiPPO-Rand | 69.93 |
| LMU | 97.08 |
| ExpRNN | 94.67 |
| GRU | 93.04 |
| LSTM | 92.54 |
| MGU | 89.37 |
| RNN | 52.98 |

Table 5: **Comparison to prior methods for pixel-by-pixel image classification.** Reported test accuracies from previous works on pixel-by-pixel image classification benchmarks. Top: Our methods. Middle: Recurrent baselines and variants. Bottom: Non-recurrent sequence models with global receptive field.

| Model | Test accuracy (%) |
|---|---|
| **HiPPO-LegS** | **98.3** |
| HiPPO-Laguerre | 98.24 |
| HiPPO-LegT | 98.03 |
| LMU (ours) | 97.29 |
| URLSTM + Zoneout [46] | 97.58 |
| LMU [71] | 97.15 |
| URLSTM [31] | 96.96 |
| IndRNN [49] | 96.0 |
| Dilated RNN [10] | 96.1 |
| r-LSTM [69] | 95.2 |
| LSTM [31] | 95.11 |
| TrellisNet [6] | 98.13 |
| Temporal ConvNet [5] | 97.2 |
| Transformer [69] | 97.9 |

## F.3   Copying

**Task**   In the Copying task [3], the input is a sequence of $L + 20$ digits where the first 10 tokens $(a_0, a_1, ..., a_9)$ are randomly chosen from $\{1, ..., 8\}$, the middle N tokens are set to 0, and the last ten tokens are 9. The goal of the recurrent model is to output $(a_0, ..., a_9)$ in order on the last 10 time steps, whenever the cue token 9 is presented. Models are trained using the cross-entropy loss; the random guessing baseline has loss $\log(8) \approx 2.08$. We use length $L = 200$. The training and testing examples are generated in the same way.

Our motivation of studying the Copying task is that standard models such as the LSTM struggle to solve it. We note that the Copying task is much harder than other memory benchmarks such as the Adding task [3], and we do not consider those.

Figure 7: Loss on the Copying task. HiPPO methods are the only to fully solve the task. The hyperparameter-free LegS update is best, while methods with timescale parameters (e.g. LegT) do not solve the task if mis-specified.

**Results**   The HiPPO-LegS method solves this task the fastest. The LegT method also solves this task quickly, only if the parameter $\theta$ is initialized to the correct value of 200. Mis-specifying this timescale hyperparameter to $\theta = 20$ or $\theta = 2000$ drastically slows down the convergence of HiPPO-LegT. The LMU (at optimal parameter $\theta = 200$) solves this task at comparable speed; like in Appendix F.2, differences between HiPPO-LegT ($\theta = 200$) and LMU here arise from the minor architecture difference in Appendix F.1.

The HiPPO-Rand baseline (denoted "random LTI" system here) does much worse than the updates with the dynamics derived from our framework, highlighting the importance of the precise dynamics (in contrast to just the architecture).

Standard methods such as the RNN and LSTM are also nearly stuck at baseline.

### F.4   Trajectory Classification

**Dataset**   The Character Trajectories dataset [4] from the UCI machine learning repository [25] consists of pen tip trajectories recorded from writing individual characters. The trajectories were captured at 200Hz and data was normalized and smoothed. Input is 3-dimensional ($x$ and $y$ positions, and pen tip force), and there are 20 possible outputs (number of classes). Models are trained using the cross-entropy loss. The dataset contains 2858 time series. The length of the sequences is variable, ranging up to 182. We use a train-val-test split of 70%-15%-15%.

**Methods**   RNN baselines include the LSTM [34], GRU [17], and LMU [71]. Our implementations of these used 256 hidden units each.

The GRU-D [11] is a method for handling missing values in time series that computes a decay between observations. The ODE-RNN [61] and Neural CDE (NCDE) [40] baselines are state-of-the-art neural ODE methods, also designed to handle irregularly-sampled time series. Our GRU-D, ODE-RNN, and Neural CDE baselines used code from Kidger et al. [40], inheriting the hyperparameters for those methods.

All methods trained for 100 epochs.

**Timescale mis-specification**   The goal of this experiment is to investigate the performance of models when the timescale is mis-specified between train and evaluation time, leading to distribution shift. We considered the following two standard types of time series:

1. Sequences sampled at a fixed rate
2. Irregularly-sampled time series (i.e., missing values) with timestamps

Timescale shift is emulated in the corresponding ways, which can be interpreted as different sampling rates or trajectory speeds.

1. Either the train or evaluation sequences are downsampled by a factor of 2
2. The train or evaluation timestamps are halved.[9]

The first scenario in each corresponds to the original sequence being sampled at 100Hz instead of 200Hz; alternatively, it is equivalent to the writer drawing twice as fast. Thus, these scenarios correspond to a train $\rightarrow$ evaluation timescale shift of 100Hz $\rightarrow$ 200Hz and 200Hz $\rightarrow$ 100Hz respectively.

Note that models are unable to obviously tell that there is timescale shift. For example, in the first scenario, shorter or longer sequences can be attributed to the variability of sequence lengths in the original dataset. In the second scenario, the timestamps have different distributions, but this can correspond to different rates of missing data, which the baselines for irregularly-sampled data are able to address.

### F.5 Online Function Approximation and Speed Benchmark

**Task** The task is to reconstruct an input function (as a discrete sequence) based on some hidden state produced after the model has traversed the input function. This is the same problem setup as in Section 2.1; the online approximation and reconstruction details are in Appendix C. The input function is randomly sampled from a continuous-time band-limited white noise process, with length $10^6$. The sampling step size is $\Delta t = 10^{-4}$, and the signal band limit is 1Hz.

**Models** We compare HiPPO-LegS, LMU, and LSTM. The HiPPO-LegS and LMU model only consists of the memory update and not the additional RNN architecture. The function is reconstructed from the coefficients using the formula in Appendix D, so no training is required. For LSTM, we use a linear decoder to reconstruct the function from the LSTM hidden states and cell states, trained on a collection of 100 sequences. All models use $N = 256$ hidden units. The LSTM uses the $L2$ loss. The HiPPO methods including LMU follow the fixed dynamics of Theorem 1 and Theorem 2.

**Speed benchmark** We measure the inference time of HiPPO-LegS, LMU, and LSTM, in single-threaded mode on a server Intel Xeon CPU E5-2690 v4 at 2.60GHz.

### F.6 Sentiment Classification on the IMDB Movie Review Dataset

**Dataset** The IMDB movie review dataset [50] is a standard binary sentiment classification task containing 25000 train and test sequences, with sequence lengths ranging from hundreds to thousands of steps. The task is to classify the sentiment of each movie review into either positive or negative. We use 10% of the standard training set as validation set.

**Methods** RNN baselines include the LSTM [34], vanilla RNN, LMU [71], and expRNN [48]. Our implementations of these used 256 hidden units each.

**Result** As shown in Table 6, our HiPPO-RNNs have similar and consistent performance, on par or better than LSTM. Other long-range memory RNN approaches that constrains the expressivity of the network (e.g. expRNN) performs worse on this more generic task.

### F.7 Mackey Glass prediction

The Mackey-Glass data [52] is a time series prediction task for modeling chaotic dynamical systems. We build on the implementation of Voelker et al. [71]. The data is a sequence of one-dimensional observations, and models are tasked with predicting 15 time steps into the future. The models are 4-layer stacked recurrent neural networks, trained with the mean squared error (MSE) loss. Voelker et al. [71] additionally consider a hybrid model with alternating LSTM and LMU layers, which improved on either by itself. We did not try this approach with our method HiPPO-LegS such as combining it with the LSTM or other HiPPO methods, but such ideas could further improve our performance. As a baseline method, the identity function does not simulate the dynamics, and simply guesses that the future time step is equal to the current input.

| Model | Test accuracy (%) |
|---|---|
| HiPPO-LegS | $87.8 \pm 0.2$ |
| HiPPO-LagT | $\mathbf{88.0} \pm 0.2$ |
| HiPPO-LegT $\theta = 100$ | $87.4 \pm 0.3$ |
| HiPPO-LegT $\theta = 1000$ | $87.7 \pm 0.2$ |
| HiPPO-LegT $\theta = 10000$ | $87.9 \pm 0.3$ |
| HiPPO-Rand | $82.9 \pm 0.3$ |
| LMU $\theta = 1000$ | $87.7 \pm 0.1$ |
| LSTM | $87.3 \pm 0.4$ |
| expRNN | $84.3 \pm 0.3$ |
| RNN | $67.4 \pm 7.7$ |

Table 6: IMDB test accuracy, averaged over 3 seeds. Top: Our methods. Bottom: Recurrent baselines.

Fig. 8 plots the training and validation mean squared errors (MSE) of these methods. The table reports final normalized root mean squared errors (NRMSE) $\sqrt{\frac{\mathbb{E}\left[(Y - \hat{Y})^2\right]}{\mathbb{E}[Y^2]}}$ between the targets $Y$ and predictions $\hat{Y}$. HiPPO-LegS outperforms the LSTM, LMU, and the best hybrid LSTM+LMU model from [68], reducing normalized MSE by over 30%.

| Model | Test MSE | Test NRMSE |
|---|---|---|
| Baseline | 0.1229 | 1.62274 |
| LSTM | 4.784e-4 | 0.10123 |
| LMU | 4.414e-4 | 0.09722 |
| Hybrid LSTM+LMU | 2.198e-4 | 0.06862 |
| LegS | 1.054e-4 | 0.04752 |

Figure 8: Mackey-Glass predictions

## F.8 Additional Analysis and Ablations of HiPPO

To further analyze the tradeoffs of the memory updates derived from our framework, in Fig. 9 we plot a simple input function $f(x) = 1/4 \sin x + 1/2 \sin(x/3) + \sin(x/7)$ to be approximated. The function is subsampled on the range $x \in [0, 100]$, creating a sequence of length 1000. This function is simpler than the functions sampled from white noise signals described in Appendix F.5. Given this function, we use the same methodology as in Appendix F.5 for processing the function online and then reconstructing it at the end.

In Figure 9(a, b), we plot the true function $f$, and its absolute approximation error based on LegT, LagT, and LegS. LegS has the lowest approximation error, while LegT and LagT are similar and slightly worse than LegS. Next, we analyze some qualitative behaviors.

**LegT Window Length** In Figure 9(c), shows that the approximation error of LegT is sensitive to the hyperparameter $\theta$, the length of the window. Specifying $\theta$ to be even slightly too small (by $0.5\%$ relative to the total sequence length) causes huge errors in approximation. This is expected by the HiPPO framework, as the final measure $\mu^{(t)}$ is not supported everywhere, so the projection problem does not care that the reconstructed function is highly inaccurate near $x = 0$.

**Generalized LagT Family** Our LagT method actually comprises a family of related transforms, governed by two parameters $\alpha, \beta$ specifying the original measure and the tilting (Appendix D.2). Fig. 10 shows the error as these parameters change. Fig. 10(a) shows that small $\alpha$ generally performs better.

Fig. 10(b, c) show that the reconstruction is unstable for larger $\beta$, but small values of $\beta$ work well. More detailed theoretical analysis explaining these tradeoffs would be an interesting question to analyze.

**LegS vs. LegT** In comparison to LegT, LegS does not need any hyperparameters governing the timescale. However, suppose that the LegT $\theta$ window size was chosen perfectly to match the length of the sequence; that is, $\theta = T$ where $T$ is the final time range. Note that at the end of consuming the input function (time $t = T$), the measures $\mu^{(t)}$ for LegS and LegT are *both* equal to $\frac{1}{T}\mathbb{I}_{[0,T]}$ (Sections 2.3 and 3). Therefore, the approximation $\text{proj}_T(f)$ is specifying the same function for both LegS and LegT at time $t = T$. The sole difference is that LegT has an additional approximation term for $f(t - \theta)$ while calculating the update at every time $t$ (see Appendix D.1), due to the nature of the sliding rather than scaling window.

(a) True function $f(x)$     (b) Absolute approx. error     (c) Error for different $\theta$'s in LegT

Figure 9: Function approximation comparison between LegT, LagT, and LegS. LegS has the lowest approximation error. LegT error is sensitive to the choice of window length $\theta$, especially if $\theta$ is smaller than the length of the true function.

(a) Generalized Laguerre family, fixed $\beta = 0.01$ and varying $\alpha$

(b) Generalized Laguerre family, fixed $\alpha = 0$ and small $\beta$

(c) Generalized Laguerre family, fixed $\alpha = 0$ and large $\beta$

Figure 10: Function approximation comparison between different instantiations of the generalized tilted Laguerre family (Appendix D.2).