[Reviews · NeurIPS 2020]

Review 1

Summary and Contributions: POST REBUTTAL The authors' rebuttal is acceptable and my score remains as it is. +++++++ The authors describe a novel framework, building on established methods, which tackles the problem of online function approximation in the presence of sequentially introduced data. This problem is framed as a memory problem to learn sequential dependencies in data to perform a given task. The main contribution of this work is to leverage tools from signal processing theory whereby function approximation using orthogonal polynomial bases are formulated with respect to time dependent measures on the input domain. The authors present an online algorithm based on ordinary differential equations that govern time-dependent projection coefficients which minimize a measure-specific L2 norm. A complete theoretical description of the algorithm and its approximation guarantees is presented, and a number of numerical experiments comparing the method to popular RNN architectures is conducted.

Strengths: The paper presents a strong and clear theory that unifies past work using orthogonal function bases to approximate sequential dependencies in machine learning. The advantages of this method are well explained and supported by theorems and experiments. While this work build on a large body of work from signal processing and approximation theory, as well as recent work from the ML community (namely Legendre Memory Units), it provides important generalizations and novel instances in the form of scaled Legendre approximations which show interesting results. I believe this paper presents significant novel results and is relevant to the NeurIPS community.

Weaknesses: An important point of comparison that is missing in the paper is a clear discussion about the role of cost functions in the comparison of RNN baselines with the HiPPO methods. If I understand correctly, HiPPO computes optimal polynomial coefficients with respect to a measure-specific L2 norm. However, it is not clear if the cost function used by the baseline comparison models was the same L2 norm, or something different. Moreover, it is not clear how this method could be adapted to cases where loss cannot be expressed with an L2 norm. A discussion about this important point is lacking. For example, in the sequential MNIST case, cross entropy loss is often used but it is not clear what is implemented here by HiPPO nor the other RNN baselines.

Correctness: All claims seem correct to me. See additional feedback comment for minor suggestions and questions about some derivations.

Clarity: The paper is very well written.

Relation to Prior Work: Prior work is clearly described, and relation between distinct lines of related work is well articulated.

Reproducibility: No

Additional Feedback: About reproducibility: have some concerns about experiments. Perhaps I missed it but cost functions used to train RNN baselines are not specified. This relates to the major weakness I described above. This is easily fixed and I am expecting to change my answer about reproducibility upon seeing this information. Here are other minor suggestions that would improve the paper: - A discussion about cost functions for RNNs in relation to HiPPO. - For section starting at line 89, a clarification that the function space should be of integrable functions would add formalism. - (Important) Line 132: it is stated that the coefficients c(t) follow a particular form of ODE. This statement appears in a definition, but it is highly non trivial and depends on involved derivations found in the appendix. To improve reading, this statement should not be part of a definition, but rather should be presented as a consequence of the formulation. A clear reference to the appendix where the derivation takes place should be added. - Details about network size and polynomial order N in experiments should be brought to the main text. These are important comparison points and are now buried in the appendix. - Appendix B.2: I believe there is a mistake in the statement of the general Leibniz Integral Rule. Multipliers d\alpha/dt and d\beta/dt should be added to the f(\alpha(t),t) and f(\beta(t),t) terms respectively. - In appendix F, line 1095. In "HiPPO variants tie the memory size N to the hidden state dimension d.", what does "tie" mean. Does it mean equal ? This is an important point of comparison and should be made clear. When the N for RNNs is specified in experiments, should we assume that the order of polynomials is the same ? - line 1137: Is the N used to designate the length of input sequences the same as the polynomial order ? If yes, this is problematic. If no, then another variable name should be used.


Review 2

Summary and Contributions: This paper develops a general framework called HIPPO for sequential data learning. Its core idea is that projecting continuous signals or discrete time series onto orthogonal polynomial bases in an online approximation way. One of its variants HiPPO-LegS present the advantages of the scaled measure and it can efficiently handle dependencies across millions of time steps.

Strengths: This framework provides an unified perspective on memory mechanisms and it can be incorporated into existing recurrent models such that all history can be remembered.

Weaknesses: Since recurrent neural networks have been studied as universal approximators for modeling dynamic systems, can you discuss more about the relationship between the HiPPO online approximation framework and recurrent networks? Is it possible directly using HiPPO to fit timeseries in the input space? It is not clear which complex dependencies in the hidden space of RNN need to be approximated by polynomial approximation. Can these fitted dependencies be visualized? It would be interesting to discuss more about the memory dynamics learned/approximated by HiPPO. Since the HiPPO can easily fit long-range function, how to adapt to short-term time series forecasting tasks? Please discuss what types of time series tasks the HiPPO can do well and which ones it can’t work well. How to find the number of polynomial bases?

Correctness: This work is studied based on the approximation theory. Details for each part are provided clearly in appendix.

Clarity: Yes, this work is well written.

Relation to Prior Work: Yes, related work is well covered.

Reproducibility: Yes

Additional Feedback: I thank the authors for their response, and am maintaining my score.


Review 3

Summary and Contributions: The paper presents a general framework (HiPPO) for the online compression of continuous signals and discrete time series by projection onto polynomial bases. The authors introduced the proposed HiPPO memory in RNN based models and evaluate the effectiveness of their models on the MNIST classification task and on a character classification task.

Strengths: The paper is well written and the idea is novel. The appendices give enough information to let the reader to follow the paper in good conditions. The paper also give different examples for a non expert reader.

Weaknesses: The paper needs more experiments to better support the proposed model. Indeed, the authors list different AI related area potentially interested by such as novel memory management in RNN based models such as speech processing, NLP, etc.

Correctness: The paper seems correct. I have not checked all equations and indices.

Clarity: Yes

Relation to Prior Work: Yes, but some comparaison to other RNN based cells that manage memory in an efficient way will be interesting such as MemRNN, Minimal RNN, PMU, etc.

Reproducibility: Yes

Additional Feedback:


Review 4

Summary and Contributions: Update after author feedback: The authors addressed my concerns and I'll raise my score to 8. _________________________________________________________________ This paper proposes a framework for compressing and representing the cumulative history of a continuous or discrete signal. The representation is obtained as coefficients by formulating the problem as optimal function approximation under certain base and measure. The dynamics of the coefficients are modeled as an ODE which can be efficiently computed with fixed matrices.

Strengths: The proposed representation can be efficiently computed and combined into conventional RNN framework. With proper measure, the representation is able to capture and recover long history due to its nature of function approximation. The modeled dynamic is able to handle certain distribution shift like unmatched sampling rate of training and testing data. It also provides a general view that unify some previous recurrent structures like vanilla LSTM and GRU as well as LMU.

Weaknesses: The performance may rely on the measure and task. The tasks in the experiments seem to favor a measure that faithfully and uniformly memorize the history like LegS, while it remains to be seen for tasks where certain local segments of history plays a more important role.

Correctness: For the speed comparison with LMU, as the dynamic is mostly similar for the representation, why is there a large gap between the proposed method and LMU?

Clarity: The paper is well written.

Relation to Prior Work: It might be better if the authors could discuss more about the speed difference between the proposed method and LMU.

Reproducibility: Yes

Additional Feedback:

[Author Response · NeurIPS 2020]

We thank all reviewers for their thoughtful feedback and suggestions. We are encouraged that all reviewers found the paper well-written and agreed that the technical ideas are important and novel. Based on suggestions from the reviewers, we have further improved the clarity of the paper and included additional details.

**(R1) - Interaction between HiPPO and cost function.** The overall HiPPO-RNN model (Sec. 4) uses the same loss function as the baselines (e.g. cross-entropy for classification, as in pMNIST). Note that the HiPPO-RNN cell in Fig. 2 depicts only the recurrent dynamics; as is convention, each cell also outputs the state $h_t$ for another layer (e.g. softmax) to compute the final prediction. When incorporated in an end-to-end model such as an RNN, the HiPPO module (blue circle in Fig. 2) can simply be viewed as a fixed, non-trainable component – although it has the additional interpretation of memorizing the history of features (in L2 space). Since it only affects feature representation, the HiPPO component does not impact the choice of output space and loss function, which is inherited from the base RNN. These details will be clarified in the next version of the paper, and are also explicit in the submitted supplementary code which will be released.

**(R1) - Reproducibility and experiment details.** Additional space permitted, we have moved much of the architecture details in Appendix F.1 to the main body, including (*) a better diagram of the HiPPO-RNN cell, (*) a discussion of the output layer and loss function, (*) a comparison of network architecture, size, and hyperparameters against all RNN baselines (in summary, they are controlled to be all roughly equal), (*) loss function details for all tasks in Appendix F.

**(R1) - Technical details.** We appreciate R1's close reading of the paper and suggestions for improving the technical presentation. We agree with all details on integrable functions, the Leibniz rule, the Copying task hyperparameter, and the interpretation of coefficients $c(t)$. Thanks for the suggestions and corrections!

**(R1, R2) - Finding the polynomial order $N$.** In the setting of approximating smooth functions, $N$ can be calculated from the explicit error bounds (Sec. 3, Prop. 6). In our end-to-end experiments, $N$ is a hyperparameter analogous to the hidden dimension of standard RNN/LSTM models; we set $N$ equal to the hidden dim. $d$ of the baselines, so that all HiPPO methods have the same number of hidden units and the same (or smaller) model size as the LSTM baseline (line 1094-1104).

**(R2) - HiPPO, RNNs, and universal approximators for dynamical systems.** Similar to universal approximation theorems, HiPPO formalizes the ability of RNNs to approximate functions. HiPPO shows that optimal compression of signals is governed by linear recurrent systems (Thm. 1, 2). Unlike universal approximation theorems, HiPPO shows non-asymptotic error rates on the approximation in terms of memory capacity (Prop. 6). The HiPPO framework is a bridge between RNNs and dynamical systems, and explains how to derive the form of a modern RNN update from first principles.

**(R2) - Directly fitting timeseries in the input space.** HiPPO can be used directly on input features, which is how the basic HiPPO framework is described. For example, the function approximation experiments (Sec. 4.3 and App. F.8) can be interpreted as fitting a long input timeseries with a budget of 256 features. We remark that the presented figures (Fig. 3, 9, 10) are a good visual indicator of the approximation dynamics of HiPPO.

**(R2, R4) - Short-term or more local time series tasks.** While HiPPO was designed to capture long-range dependencies, several of our experiments involve tasks requiring shorter memory: trajectory classification in Sec. 4.2 (length $\approx 100$), chaotic dynamics simulation in App. F.7 (length 15), and the IMDB dataset in App. F.6 (some inputs of length $< 100$).

**(R3) - Experiments.** R3's only concern is that "the paper needs more experiments to better support the proposed model... such as speech processing, NLP, etc." Although we acknowledge R3's suggestion, we note that the submission already builds up from the foundational theory to a comprehensive range of experiments including an NLP application. Indeed, in addition to directly validating the proposed model and online function approximation theory on synthetic experiments, we have included experiments on memory benchmarks, image classification, trajectory classification, timescale robustness, chaotic dynamics prediction, and NLP (the IMDB movie review benchmark); some of these are in Appendix F due to space constraints. We are excited about pursuing further applications in follow-up work.

**(R3) - Baselines.** Although R3 asks about more memory RNN baselines, the memory experiments in the paper include a variety of well-known memory models (including RNNs), chosen for their relevance to each task. Our permuted MNIST results explicitly compare against 15 sequence model baselines (Tab. 4, 5) to establish SoTA; this also establishes a large number of implicit comparisons, e.g., the LMU outperformed 8 other memory RNN models in (Voelker 2019). We ran an additional baseline suggested by R3, the MinimalRNN, on the memory benchmarks (Sec. 4.1). This model failed to solve the Copying task and achieved 89.1 accuracy on the permuted MNIST benchmark, compared to the SoTA 98.3% of our model.

**(R4) - Why LegS is faster than LMU.** The speed difference in Sec. 4.3 is explained by our faster algorithm for LegS. Because of the approximation made by LMU due to its measure (Sec. 2.3 lines 158-161), the LMU transition matrix $A$ (Thm. 1) is not triangular, in contrast to that of LegS (Thm. 2). Numerically stable discretizations (App. B.3) require inverting this matrix, which is efficient in the LegS case (Prop. 4, proved in App. E.2) but not known for the LMU case.

**(R4) - Performance depends on measure and task.** We fully agree with R4's observation that some tasks are more suited to LegS due to the uniform memorization prior. In fact, rather than seeing this as a weakness, we argue that a primary contribution of this paper is introducing the technical framework that exposes these very tradeoffs. This framework allowed us to (1) explain the memory mechanism of existing methods (LMU, LSTM, Fourier Recurrent Unit, etc.) in terms of their approximation measures, (2) introduce new methods that may be more appropriate in different settings such as very long memory and mis-specified timescales, and (3) theoretically analyze and contrast different approaches based on their underlying mechanisms. We thank R4 for bringing up this insightful point!

[Meta-Review · NeurIPS 2020]

This is a novel and interesting piece of work. I encourage the authors to take the reviewers comments into account when revising the manuscript: the experiments in the main text are somewhat small scale and there were some comments about how the text could be improved.